

**Distribution and behaviour of dissolved selenium in**
**tropical peatland-draining rivers and estuaries of**
**Malaysia**
Yan Chang[1*], Moritz Müller[2], Ying Wu[1], Shan Jiang[1], Wan wan Cao[1], Jian guo
Qu[1], Jing ling Ren[3], Xiaona Wang [1], En ming Rao[3], Xiao lu Wang[1], Aazani
Mujahid[4], Mohd Fakharuddin Muhamad[2], Edwin Sia Sien Aun[2], Faddrine Holt
Ajon Jang[2], Jing Zhang[1]
[1] State Key Laboratory of Estuarine and Coastal Research, East China Normal
University, Shanghai 200062, China
[2] Faculty of Engineering, Computing and Science Swinburne, University of
Technology, Sarawak 93350, Malaysia
[3] Key Laboratory of Marine Chemistry Theory and Technology, Ministry of
Education, Ocean University of China, Qingdao 266100, China
[4] Faculty of Resource Science and Technology, University Malaysia Sarawak,
Sarawak 93350, Malaysia
Correspondence to:
Yan Chang
Email: ychang@sklec.ecnu.edu.cn



## Abstract

Selenium (Se) is an essential micronutrient for many organisms. Despite
its importance, our current knowledge of the biogeochemical cycling of
dissolved Se in tropical estuaries is limited, especially in Southeast Asia. To
gain insights into Se cycling in tropical peat-draining rivers and estuaries,
samples were collected from the Rajang, Maludam, Sebuyau, Simunjan,
Sematan, Samunsam, and Lunda rivers and estuaries in western Sarawak,
Malaysia, in March and September 2017 and analysed for various forms of Se
(dissolved inorganic and organic). Mean total dissolved Se (TDSe), dissolved
inorganic Se (DISe), and dissolved organic Se concentrations (DOSe) were 2.2
nmol $L^{-1}$ (range: 0.7 to 5.7 nmol $L^{-1}$), 0.18 nmol $L^{-1}$ (range: less than the
detection limit to 0.47 nmol $L^{-1}$), and 2.0 nmol $L^{-1}$ (range: 0.42 to 5.7 nmol $L^{-1}$),
respectively. In acidic, low-oxygen, organic-rich blackwater (peatland-draining)
rivers, the concentrations of DISe were extremely low, whereas those of DOSe
were high. In rivers and estuaries that drained peatland, DOSe/TDSe ratios
ranged from 0.67 to 0.99, showing that DOSe dominated. The positive
relationship between DISe and salinity and the negative relationship between
DOSe and salinity indicate marine and terrestrial origins of DISe and DOSe,
respectively. The positive correlations of DOSe with the humification index and
humic-like chromophoric dissolved organic matter components in freshwater
river reaches suggest that peat soils are probably the main source of DOSe.
Discharges of water enriched with DOSe fractions associated with peatland-
derived high-molecular-weight, high-aromaticity dissolved organic matter
discharged from estuaries may promote productivity in the adjoining
oligotrophic coastal waters. The results of this study suggest that the impacts
of Se discharges on coastal ecosystems should be evaluated in the future.





## 1. Introduction


Selenium (Se) is an essential trace element for animals and most plants.
Low levels of Se in the food chain lead to disease or death (Lobanov et al.,
2009; Winkel et al., 2015), whereas high levels are toxic. The range of beneficial
effects of Se is among the narrowest of all the elements and varies between
dietary deficiency (<40 µg d$^{-1}$) and toxicity (>400 µg d$^{-1}$) (Fernández-Martínez
and Charlet 2009; Schiavon et al., 2017). Selenium depletion in the
Phanerozoic oceans may have contributed to three major mass extinction
scenarios (Long et al., 2016). Thus, there has been interest in Se
biogeochemical cycling in aquatic systems for many decades (e.g., Cutter and
Bruland, 1984; Cutter and Cutter, 1995, 2001; Mason et al., 2018).
The bioavailability of Se is determined by its concentrations and species
(Fernandez and Charlet, 2009). The behaviour of selenium in natural waters is
complicated, as it exists in several oxidation states (−II, IV, VI) and as organic
selenide (Conde and Sanz Alaejos 1997). A number of field and laboratory
studies have found that selenite [Se(IV)] and selenate [Se(VI)] can be
assimilated by phytoplankton and that Se(IV) is the preferred species for marine
phytoplankton (Wrench and Measures, 1982; Apte et al., 1986; Vandermeulen
and Foda, 1988; Baines and Fisher, 2001). Substantial amounts of dissolved
Se in natural waters are known to be associated with organic matter, including
water-soluble proteins, polysaccharides, and humic substances (Ferri and
Sangiorgio,1999; Cutter and Cutter, 1995, 2001; Kamei-Ishikawa et al., 2008),
with the bioavailability of Se generally decreasing as the amount of organic
matter increases (De Temmerman et al., 2014; Winkel et al., 2015). Se(IV),
when added to raw humus layers in a forest, was found in a field study to be
fixed very rapidly (Gustafsson and Johnsson, 1992, 1994). Laboratory studies
have shown that Se(IV) is adsorbed by peat (Kharkar et al., 1968) and that Se



is accumulated and stored in dome-shaped peat swamps (Gonzalez et al., 2006,
Vesper et al., 2008; Clark and Johnson, 2008). In a global study, Fernández-
Martínez and Charlet (2009) summarized that the concentrations of Se in soils
generally ranged from about 0.01 to 2 mg kg$^{-1}$ and averaged about 0.44 mg
kg$^{-1}$. Gonzalez et al. (2006) reported Se concentrations of up to 28 mg kg$^{-1}$ in
peatland in Switzerland and from 0.9 to 2.2 mg kg$^{-1}$ in peat cores in Spain. High
spatial variability has been found in dissolved Se concentrations in runoff from
peatlands at regional scales, with concentrations of up to 13 nmol L$^{-1}$ being
observed in northern Minnesota, US (Clausen and Brooks, 1983), and from
0.38 to 5 nmol L$^{-1}$ in the Krycklan catchment, Sweden (Lidman et al., 2011).
Although these various studies did not report different species of Se, the organic
form of Se is probably more important than inorganic forms in runoff from
peatland. An understanding of Se speciation may therefore be important for
determining the bioavailability of Se that is transported from land to oceans.

The chemical behaviour of Se in estuarine mixing plays an important role

in overall geochemical cycling. From their investigation into dissolved Se
species in various estuaries, Chang et al. (2016) found that Se was controlled
by biological, physical, and redox processes in the estuaries; non-conservative
processes resulting from phytoplankton uptake; absorption by suspended
particles; and regeneration of particulate organic selenide in the water. Thus far,
the behaviour of Se in estuaries has been studied mainly in the temperate zone
of the northern hemisphere (between 20°N and 60°N) (Measures and Burton,
1978; Takayanagi and Wong, 1984; Van der Sloot et al., 1985; Cutter, 1989a;
Guan and Martin, 1991; Hung and Shy, 1995; Abdel-Moati, 1998; Yao et al.,
2006; Chang et al., 2016). Wide spatial and temporal variations have been
reported in total dissolved Se concentrations in runoff from high-latitude
peatlands (Clausen and Brooks, 1983; Lidman et al., 2011). The behaviour of
Se in tropical organic-rich estuaries, however, is still poorly understood. It is



also known that organic matter plays an important role in the bioavailability and
fate of Se in the environment; for example, Moore et al. (2013) and Wit et al.
(2015) reported very high concentrations (up to 5667 µmol L$^{-1}$) of dissolved
organic carbon (DOC) in peat-draining rivers in Borneo. More studies of the
behaviour of Se in fluvial systems in Southeast Asia are therefore needed to
provide an improved understanding of the biogeochemical processing of Se
fractions and their relationships with organic matter.
To the best of our knowledge, the present study is the first analysis of the
distribution and behaviour of dissolved species of Se in seven rivers and
estuaries in western Borneo (Sarawak, Malaysia, Southeast Asia). The main
objectives of the study were to 1) investigate and compare the distribution of
dissolved Se species, including dissolved inorganic Se [DISe, the sum of Se(IV)
and Se(VI)] and dissolved organic Se (DOSe) along salinity gradients in rivers
with high (Maludam, Simunjan, Sebuyau, and Samunsam) and limited (Rajang,
Semetan, and Lundu) proportions of peatland in the wet and dry seasons; 2)
evaluate the fate of Se species in multiple estuaries during the mixing of
freshwater and salt water in different seasons; and 3) characterize the DOSe
fractions in the peat-draining rivers and estuaries. The results of this study
should contribute to an improved understanding of how Se behaves in tropical
peat-draining rivers and estuaries.

## 2. Materials and methods

### 2.1 Study areas and sample collection

Sarawak, Malaysia's largest state, is in the northwest of the island of
Borneo, Malaysia (Müller et al., 2016). The coastline of Sarawak is about 1035
km long, and the offshore comprises a wide continental shelf area with high



biological productivity (Long et al., 2014). Sarawak has a tropical climate, with
a mean annual air temperature at the capital Kuching (1.56°N, 110.35°E) of
26.1 °C (Müller et al., 2016). Rainfall is abundant throughout the year but is
pronounced during the northeastern monsoon, which occurs between
November and February (wet season). The period from May to September,
before the southwestern monsoon, constitutes the dry season (Sa'adi et al.,
2017). About 12% of the coastal area of western Sarawak is covered by
peatlands, of which approximately 41% has been converted to palm plantations
(Müller et al., 2016).

Two sampling campaigns were conducted in peat-draining rivers and

estuaries in Sarawak in 2017. The first was at the end of the northeastern
monsoon (from 4 to 12 March 2017, just after the wet season), and the second
was shortly before the beginning of the southwestern monsoon (from 4 to 17
September 2017, in the dry season) (Figure 1). Six rivers, namely, the Rajang,
Maludam, Simunjan, Sebuyau, Sematan, and Samunsam, were sampled in
March and September, and the Lundu River was sampled only in September
(Fig. 1). Four of the rivers (the Maludam, Simunjan, Sebuyau, and Samunsam)
drain catchments with high peatland coverages and are known as blackwater
rivers, whereas the Sematan and Lundu drain catchments with high proportions
of mineral soils and limited proportions of peatlands (Martin et al., 2018). The
Rajang River drains mineral soils in its upper reaches (Staub et al., 2000) but,
at Sibu, branches into multiple distributary channels (the Igan, Paloh, and
Rajang) that flow from north to south through land covered with thick peat and
form a delta (Staub et al., 1994, 2000) (Fig. 1).

Water samples were collected from a boat. As the boat moved forward,

surface water was collected upstream and to the side of the boat into an acid-
cleaned polyethylene bottle attached to the end of a plastic pole sampler (3–4
m long). Water temperature, salinity, pH, and dissolved oxygen (DO)





concentrations were measured *in situ* using a portable multifunction water-
quality meter (AP–2000, Aquaread Company, Britain) at the time of sample
collection. Water samples were filtered within 12 h of collection through pre-
cleaned 0.4 μm filters (Nuclepore) at a laminar air flow cleanbench (Class 100).
The filtrates were placed in acid-cleaned polyethylene bottles and were frozen
and stored until analysis.
**2.2  Mixing experiment**

To supplement the field observations, a laboratory experiment that

simulated estuarine mixing processes was carried out using freshwater
collected from the Maludam (organic rich and yellow coloured, with humic
substances) and Rajang rivers during September 2017. Samples of freshwater
(salinity of 0) were collected from the Maludam River at Maludam National Park
and from the Rajang River at Sibu (10 km downstream from the city dock). The
dissolved organic carbon (DOC) concentrations of these samples were 121 and
3631 μmol L$^{-1}$ (Martin et al., 2018), respectively. Coastal seawater with a salinity
of 32 and a DOC concentration of about 80 μmol L$^{-1}$ (Martin et al., 2018) was
also collected. The river water and the coastal seawater samples were filtered
(pre-cleaned 0.4 μm particle-free, polycarbonate membrane filters) and then
mixed at various proportions to achieve salinity gradients of 0, 8, 16, 24, and
32 (Bergquist and Boyle, 2006). Following mixing, the samples were shaken
and placed in the dark at 25–26 °C for 24 h and were then filtered through pre-
cleaned 0.4 μm polycarbonate membrane filters. The filtrates were kept frozen
until analysis.
**2.3  Analytical methods**

The Se(IV), DISe, and TDSe concentrations were determined in carbon-



containing plasma using a hydride generation (HG) system (Hydride FAST, ESI)
combined with a sector field inductively coupled plasma–mass spectrometry
(ICP–MS) instrument, as outlined in the operationally defined hydride
generation-based speciation analysis methods described by Chang et al. (2014,
2017). Selenium was measured at $m\ z^{-1}$ = 82 with low resolution. By adding
methane (2 ml min$^{-1}$) to the carbon-containing plasma, Se sensitivity was
increased and spectral interference was suppressed, which improved the
detection limits. Briefly, Se(IV) at an acidity of 2 mol l$^{-1}$ HCl was reacted with
NaBH$_4$ to produce hydrogen selenide and then quantified using HG–ICP–MS.
Se(VI) was quantitatively reduced to Se(IV) by heating a sample acidified with
3 mol l$^{-1}$ HCl to 97 °C for 75 min and then quickly cooling to room temperature
using an ice-water bath. The steps used to determine Se(IV) were then followed
to obtain the concentration of DISe. The reduction recoveries ranged from 95%
to 103%. The Se(VI) concentration was calculated as the difference between
DISe and Se(IV). The total dissolved selenium (TDSe) concentrations were
determined using the same method as for DISe, following ultraviolet digestion
(Li et al., 2014). The concentration of DOSe was calculated as the difference
between the TDSe and DISe concentrations (DOSe = TDSe − DISe). Detection
limits for Se(IV), DISe, and TDSe were 0.0025, 0.0063, and 0.0097 nmol l$^{-1}$,
respectively. The accuracy of the methods was tested with standard solutions,
and Se(IV) GSBZ 50031-94, Se(VI) GBW10032, selenocysteine GBW10087,
and selenomethionine GBW10034 showed differences within 3.0%, 0.7%, 1.6%,
and 1.4%, respectively.
**2.4 Data statistics and analysis**
The Statistical Package for Social Sciences (SPSS) version 23.0 was used
to perform Student's t-tests and linear regression analyses. The significance
level for all the analyses was $p < 0.05$.





## 3. Results

### 3.1 Water chemistry

Water temperature ranged from 26 to 32 °C throughout the study area during the two sampling periods (Table S1). In the Rajang estuary, salinity was almost 0 in the upper Igan distributary in both sampling periods, indicating strong freshwater inputs (Fig. 2). Salinity at the mouth of the Igan distributary was lower than that in the mouth of the Paloh and Rajang distributaries (Fig. 2a and 2b), reflecting the increase in tidal range from the Igan to the Rajang distributaries (Staub et al., 2000). Values of pH were lower in the riverine side, especially in the delta-plain distributaries, and increased towards the sea (Fig. S2). DO concentrations were higher in the freshwater reach than in the delta-plain distributaries in both sampling periods (Fig. S2).

In the freshwater reach of the Maludam River, pH was low (<4), and DO concentrations ranged from 1.08 to 2.4 mg $L^{-1}$; salinity, pH, and DO all increased with increasing proximity to the coast, similar to that observed for the Sebuyau, Simunjan, and Samunsam rivers in both March and September (Fig. S2). Values of pH and DO concentrations in freshwater were higher in the Samunsam than in the Maludam, Simunjan, and Sebuyau rivers, and values of pH and DO concentrations in the Sematan and Lundu, which drain mostly mineral soils, were higher than those in the blackwater rivers (Fig. S1). DO concentrations in the Simunjan River were significantly higher in September than in March ($p < 0.05$), but DO concentrations did not differ between seasons in the other rivers ($p > 0.05$). Similarly, there was no significant seasonal variation in pH in the studied rivers ($p > 0.05$).





**3.2 Se species distribution and relationship with salinity**

TDSe concentrations in the studied rivers and estuaries ranged from 1.0 to 5.7 nmol L$^{-1}$ (mean of 2.4 nmol L$^{-1}$) in March and from 0.70 to 3.9 nmol L$^{-1}$ (mean of 1.8 nmol L$^{-1}$) in September (Table S1). DISe concentrations ranged from below the detection limit (0.0063 nmol L$^{-1}$) to 0.41 nmol L$^{-1}$ (mean of 0.19 nmol L$^{-1}$) in March and from below the detection limit (0.0063 nmol L$^{-1}$) to 0.47 nmol L$^{-1}$ (mean of 0.18 nmol L$^{-1}$) in September (Table S1). DOSe concentrations ranged from 0.67 to 3.9 nmol L$^{-1}$ (mean of 1.7 nmol L$^{-1}$) in March and from 0.42 to 0.47 nmol L$^{-1}$ (mean of 0.18 nmol L$^{-1}$) in September (Table S1). DOSe/TDSe ratios ranged from 0.67 to 0.99 (mean of 0.91) and from 0.56 to 0.99 (mean of 0.88) in March and September, respectively, indicating that DOSe was the major species of Se in the peat-draining rivers and coastal estuaries in both the dry and wet seasons (Table S1).

3.2.1 Rajang estuary

In the Rajang estuary, TDSe concentrations in March and September ranged from 1.1 to 3.7 nmol L$^{-1}$ (mean of 1.9 nmol L$^{-1}$) and from 1.7 to 3.0 nmol L$^{-1}$ (mean of 2.2 nmol L$^{-1}$), respectively (Table S1). Student's t-test results showed that the concentrations of TDSe, DISe, and DOSe did not differ between the wet and dry seasons ($p > 0.05$). TDSe, DISe, DOSe, Se(IV), and Se(VI) concentrations and DOSe/TDSe ratios in the Rajang estuary are shown in Fig. 2 and S1. Se(IV) concentrations varied from 0.05 to 0.15 nmol L$^{-1}$ and were high in the coastal areas in both seasons (Fig. S1e and f). Se(IV) concentrations did not differ between the two seasons ($p > 0.05$). Se(VI) concentrations ranged from 0.068 to 0.39 nmol L$^{-1}$ and were also high in the coastal areas (Fig. S1g and h). As with Se(IV), there was limited seasonal variation in the concentrations of Se(VI). DISe concentrations reached a





maximum in the coastal areas, whereas DOSe concentrations were higher in
the delta-plain distributaries than in the upper reach in both seasons (Fig. 2c–
f). TDSe concentrations did not show a clear pattern in March but in September
were slightly higher in the delta-plain distributaries than in the upper reach (Fig.
2g and h). DOSe/TDSe ratios were high in the delta-plain distributaries and
decreased in a seaward direction to around 0.7, indicating that DOSe
dominated in the Rajang estuary (Fig. 2).

Variation in Se species concentrations along a salinity gradient in the three

tributaries (the Igan, Lassa, and Rajang) of the Rajang Estuary in March and
September are shown in Fig. 3. Theoretical mixing lines (TMLs) were
developed using two endmembers, namely, a freshwater endmember in the
freshwater reach of the Rajang River and a marine endmember with a salinity
of >30. In March, Se(IV) and Se(VI) concentrations increased with salinity and,
compared with the TML, Se(IV) and Se(VI) removals were commonly observed
in the Rajang and Paloh branches (Fig. 3a and b). In September, Se(IV) and
Se(VI) concentrations also increased with salinity, with additions of Se(IV) in the
upper reaches of the Rajang and Paloh branches and relatively little variation
in Se(VI) (Fig. 3d and e). DISe concentrations, the sum of Se(IV) and Se(VI),
increased with salinity and during mixing, and in the low-salinity water were
lower in March than in September (Fig. 3c and f). DOSe concentrations
decreased with salinity and were much higher than the TML in the Rajang and
upper Paloh branches in both March and September and in the Igan Branch in
September (Fig. 3g and j). TDSe concentrations in the mixing zone of the
Rajang and Igan branches were also higher than the TML (Fig. 3h and k).
DOSe/TDSe ratios were around 0.9 in the freshwater reach, increased to
almost 0.95 in the low-salinity water of the Igan, Paloh, and Rajang branches,
then decreased towards the sea (Fig. 3i and l).





### 3.2.2 Peat-draining rivers and estuaries


In the Maludam estuary, DISe concentrations were extremely low (near or
below the detection limits) in the freshwater reach and were around 0.3 nmol
$L^{-1}$ near the sea in both seasons (Fig. 4a). DISe concentrations followed similar
patterns in the Sebuyau, Simunjan, and Samunsam rivers and were lower in
the river than in the area closer to the sea. DISe concentrations ranged from
0.12 to 0.35 nmol $L^{-1}$ in the Sematan and Lundu and showed little seasonal
variation (Fig. 4b–e). Se(IV) and Se(VI) concentration are not presented but
were even lower than those of DISe and commonly lay below the detection limit,
especially in the freshwater reaches. In the Maludam estuary, DOSe
concentrations ranged from 1.5 to 4 nmol $L^{-1}$ and increased with distance
downstream in the freshwater area to the river mouth and then decreased
towards the sea (Fig. 4f). DOSe concentrations in the Sebuyau estuary ranged
from around 1.3 to 3.8 nmol $L^{-1}$ and followed a similar trend to those in the
Maludam estuary (Fig. 4g). In the Simunjan and Samunsam estuaries, DOSe
concentrations decreased in a seaward direction in both seasons (Fig. 4i and
j). In the Sematan and Lundu estuaries, DOSe concentrations ranged from 0.42
to 2.5 nmol $L^{-1}$, were slightly lower than those in the blackwater rivers, and
decreased in a seaward direction. DOSe/TDSe ratios were between 0.8 and
almost 1 in the freshwater reaches of the Maludam, Sebuyau, Simunjan, and
Samunsam estuaries, indicating that DOSe was the only (or dominant) species
in the freshwater of the blackwater rivers. DOSe/TDSe ratios were between 0.6
and 0.9 in the Sematan and Lundu, indicating that more than half of the Se was
still present in the form of DOSe in those rivers and estuaries with limited
peatland cover (Fig. 4l). As TDSe is the sum of the DISe and DOSe
concentrations, and DOSe generally dominated in the sampled rivers and
estuaries, the distributions of TDSe and DOSe were similar (Fig. 4m–q). TDSe,





DISe, and DOSe concentrations did not differ between seasons in the Maludam,
Sebuyau, Samunsam, Sematan or in the Rajang ($p > 0.05$). In the Simunjan
estuaries, DOSe concentrations ranged from around 1.8 to 5.7 nmol L$^{-1}$ in
March and were significantly higher than those in September ($p < 0.05$); TDSe
concentrations in this river also differed between the two seasons. The limited
seasonal variations in the Se species in the rivers and estuaries sampled in this
study may reflect the La Niña conditions that caused high precipitation and high
discharge rates in Malaysia in 2017 (Jiang et al., 2019).

Plots of DISe concentration against salinity show a positive linear

regression between DISe and salinity in the Maludam, Sebuyau, and
Samunsam estuaries ($p < 0.05$) in both seasons, but not in the Sematan estuary
($p > 0.05$), where DISe concentrations in the freshwater and marine water
endmembers were similar in both seasons (Fig. 5a–d). The salinities varied little,
either between the two seasons in the Simunjan and Lundu estuaries or in the
Sebuyau estuary in September, and therefore Se concentration–salinity
relationships were not examined. As shown in Fig. 4f and g, DOSe
concentrations in the freshwater parts of the Maludam and Sebuyau rivers
varied widely and increased downstream, so the geographical location nearest
to the river mouth with a salinity of <1 was selected as the freshwater
endmember in the linear mixing models. A negative linear correlation was
observed between DOSe concentration and salinity ($p < 0.05$) in the Maludam,
Sebuyau, and Samunsam estuaries for both seasons, but DOSe concentrations
did not vary significantly with salinity ($p > 0.05$) in the Sematan estuary (Fig.
5e–h). TDSe concentrations were also negatively correlated with salinity ($p <$
0.05) in the Maludam, Sebuyau, and Samunsam estuaries but not in the
Sematan Estuary (Fig. 5i–l). DOSe/TDSe ratios in the Maludam and
Samunsam estuaries were almost 1 when salinity was less than 10 and
decreased to around 0.8 as salinity increased. In the Sebuyau estuary, the



DOSe/TDSe ratio decreased from nearly 1 to 0.8 along the salinity gradient (Fig.
5m–o). In the Sematan estuary, DOSe/TDSe ratios remained at around 0.9
along the salinity gradient and varied widely in the coastal area in March but did
not follow any clear pattern in September (Fig. 5p).

Generally, relationships between the Se species and salinity fell into three

groups. In the blackwater estuaries (the Maludam, Sebuyau, and Samunsam),
DISe concentrations were positively correlated with salinity; DOSe and TDSe
concentrations were negatively correlated with salinity. In the Rajang estuary,
which has a large area of peatland in its delta area, DISe increased with salinity
but behaved non-conservatively and was removed in the brackish water;
whereas DOSe and TDSe decreased with salinity, behaved non-conservatively,
and were added during estuarine mixing (Fig. 3). In the Sematan estuary, TDSe,
DOSe, and DISe behaved non-conservatively and showed little change during
estuarine mixing (Fig. 5).
**3.3 Mixing experiments**

To simulate the behaviour of selenium species in different organic matter

conditions, simple mixing experiments without suspended particles were
conducted in the laboratory using water from the Rajang and Maludam
estuaries. The results of these laboratory mixing experiments are shown in Fig.
6. DISe concentrations were lower in the Maludam estuary than in the Rajang,
whereas DOSe concentrations were higher. The TML obtained when the river
and seawater components were mixed showed that when suspended particles
were excluded, there was a near-linear increase in DISe concentration with
salinity in the Rajang estuary, which indicates a marine source of DISe (Fig. 6a).
In the Maludam estuary, DISe concentrations also increased with salinity, but
the measured values were lower than the theoretical values, with removal rates
of 52% to 74%, indicating intense removal during mixing with marine water (Fig.

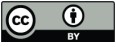



6a). In contrast to DISe, there was a near-linear decrease in DOSe
concentration with salinity in both the Rajang and Maludam estuaries, indicating
riverine sources of DOSe (Fig. 6b). In the Rajang estuary, TDSe showed a near-
linear decrease along the salinity gradient. While in the Maludam estuary, TDSe
concentrations decreased with salinity (Fig. 6c), with the measured values
being lower than the theoretical values, indicating removal processes at high
salinity (>16), mainly due to the removal of DISe (Fig. 6c). In the Maludam
estuary, DOSe/TDSe ratios ranged from nearly 1 to 0.8 in the mixing
experiments, indicating that DOSe was the major species of TDSe, with ratios
close to 1 when the salinity was less than 15, confirming the *in situ* results (Fig.
6d).

**4. Discussion**
**4.1 Se speciation in freshwater**
Considerable variation was observed in Se speciation between the studied
rivers. DISe concentrations in the blackwater rivers (Maludam, Simunjan,
Sebuyau, and Samunsam) were lower than, or close to, the detection limits
(0.0063 nmol $L^{-1}$) in the freshwater, and DOSe (from 1.3 to 5.7 nmol $L^{-1}$)
dominated TDSe in both seasons (Fig. 3). DISe concentrations were slightly
higher (from 0.12 to 0.25 nmol $L^{-1}$) and DOSe concentrations (1.0 to 2.7 nmol
$L^{-1}$) lower in the freshwater of the Rajang and Sematan rivers than in the
blackwater rivers (Fig. 7). TDSe concentrations in the sampled rivers were
comparable with those measured in other rivers worldwide (between 0.2 and
6.4 nmol $L^{-1}$); however, in contrast to our findings, DISe generally dominates in
other rivers (Cutter, 1989b; Conde and Sanz Alaejos, 1997; Pilarczyk et al.,
2019). The limited data available show that DOSe concentrations in rivers



worldwide range from <0.02 to 0.82 nmol L$^{-1}$ (Takayanagi and Wong, 1984;
Wang and Shy, 1995; Cutter and Cutter, 2001, 2004). In the blackwaters of the
Orinoco in South America, TDSe concentrations were found to range from 0.07
to 0.25 nmol L$^{-1}$ (Yee et al., 1987). Although they did not analyse DOSe
fractions directly, Yee et al. (1987) assumed that DOSe was likely to constitute
about 10%–15% of the total Se, a much lower value than the DOSe proportions
observed in peat-draining rivers in Sarawak.

The behaviour of Se in the environment is complex, as it can exist in

multiple oxidation states and as organic species (Conde and Sanz Alaejos,
1997). As shown in Fig. 7a–d, DISe concentrations were positively correlated
with the DO concentrations and pH values in the freshwaters of the studied
rivers. Se(IV)/Se(VI) ratios represent the relative proportions of Se(IV) and
Se(VI) in DISe. Se(IV)/Se(VI) ratios increased as DO concentrations and pH
values increased in March, indicating that the proportion of Se(IV) in DISe
increased as DO and pH increased. Species of Se are very sensitive to redox
conditions and pH values (Sharma et al., 2015). Sorption to solid surfaces
(including metallic oxides, hydroxides, and clays) is also a pH-dependent
process, with substantial sorption of Se(IV) and Se(VI) occurring at pH values
of 4 to 6 and negligible sorption under more alkaline conditions (pH > 8)
(BarYosef and Meek, 1987; Balistrieri and Chao, 1987; Papelis et al., 1995;
Sharma et al., 2015). Se(VI) adsorption onto solid surfaces is weaker than that
of Se(IV) (Balistrieri and Chao, 1987; Zhang and Sparks, 1990; Seby et al.,
1998). Thus, adsorption of Se(IV) and Se(VI) by metal oxyhydroxides and clays
when pH is between 4 and 6 may help to explain the low DISe concentrations
in the sampled freshwater, and DISe concentrations and Se(IV)/Se(VI) ratios
might be expected to increase as pH increases.

Peat has a high content of natural organic matter, which also plays an

important role in Se speciation (Tam et al. 1999; Li et al., 2017). Martin et al.



(2018) reported that DOC concentrations in the sampled rivers ranged from 120
to 4400 μmol L$^{-1}$. As shown in Fig. 7e, the DISe/DOSe ratio was negatively
related to DOC concentration (data from Martin et al., 2018). Almost 15% of
Se(IV) is removed by adsorption to peat (Kharkar et al., 1968). Se(IV) and Se(VI)
associated with humic and fulvic substances appear to be responsible for the
immobilization of inorganic Se (Kang et al., 1991; Zhang and Moore, 1996;
Wang and Gao, 2001), and Se sorption kinetics on humic acids can be
expressed by a pseudo-second-order equation (Kamei-Ishikawa et al., 2007).
The Maludam, Sebuyau, and Simunjan catchments are mainly peat, whereas
the Samunsam River drains an extensive area of peatland in its upper reaches
(Müller et al., 2016; Martin et al., 2018). The Rajang catchment is dominated by
mineral soils, with peatland being found only in the delta surrounding the
distributaries (Staub et al., 1994, 2000). The catchments of the Sematan and
Lundu also have limited peat deposits (Martin et al., 2018). DO, pH, and DOC
concentrations of the water probably contributed to the observed variations in
Se species, and the acidic, low-oxygen, and organic-rich blackwater rivers were
not a suitable environment for DISe.

Coal deposits in Kanawha County in the USA have been interpreted as a

dome-shaped peat swamp, analogous to those in Malaysia. Coal Se contents
reached 10.7 mg/kg, and sequential extraction results showed that the
concentrations of the organically bound fraction were the highest (Vesper et al.,
2008). It is therefore expected that organic matter that is solubilized and
leached from peat would cause Se concentrations to increase, and therefore
leaching from Se-rich peat soils is inferred to be the major source of DOSe in
our sampled rivers. A study of chromophoric dissolved organic matter (CDOM)
in these rivers and estuaries have found that humic-like CDOM components
(C1, C2, C3, and C4) were derived from peatlands (Zhou et al., 2019). DOSe
concentrations measured in the present study correlate positively with the





humification index (HIX, which represents the humification degree of dissolved
organic matter) and the sum of the humic-like CDOM components (C1, C2, C3,
and C4) ($p < 0.05$) (data from Zhou et al., 2019) in the freshwater of the studied
rivers (Fig. 7f and g). These results indicate that DOSe in Sarawak may be
associated with dissolved humic substances, which is consistent with the
findings of Zhang and Moore (1996), who reported that substantial amounts of
dissolved Se in natural waters were associated with organic matter. Gustafsson
and Johnsson (1992, 1994) found that a high proportion of the Se(IV) added to
humic lake water was adsorbed by humic substances in the form of Se(IV) to
metal–humic complexes, which is similar to phosphate adsorption by iron–
humic complexes. A study of Finnish lakes has also shown that about half of
the TDSe was present in humic substances, whereas DISe represented
between 12% and 24% of the TDSe (Wang et al., 1995). However, the
mechanisms behind the interactions between Se and dissolved organic ligands
are still poorly understood. Three hypotheses have been proposed to explain
organic-matter-mediated retention of Se, as follows: 1) direct complexation of
organic matter with Se, 2) indirect complexation via Se-cation–organic-matter
complexes, or 3) microbial reduction and incorporation into amino acids,
proteins, and natural organic matter (Winkel et al., 2015). Depending on the
type of binding, Se may be easily mobilized (e.g., through adjusting pH) or
immobilized (e.g., by covalent incorporation to organic matter) (Winkel et al.,
2015). However, there is ambiguity about the molecular structure and species
of Se that bind to organic matter, and further work is needed to identify the
mechanisms by which Se is bound to, and released from, organic matter.

**4.2 Behaviour of DISe during estuarine mixing**
As shown in Figures 3 and 5, DISe concentrations increased as salinity





increased in the Rajang, Maludam, Sebuyau, and Samunsam estuaries. These
reversed concentration–salinity relationships contrast with those reported for
other estuaries (Measures and Burton, 1978; Takayanagi and Wong, 1984; Van
der Sloot et al., 1985; Cutter, 1989a; Guan and Martin, 1991; Hung and Shy,
1995; Abdel-Moati, 1998; Yao et al., 2006; Chang et al., 2016). Laboratory
mixing experiments conducted using water from the Rajang and Maludam
rivers also revealed that DISe concentration increased as salinity increased (Fig.
6a). During estuarine mixing, DISe has been shown in some other studies to
behave conservatively ($R^2 > 0.9$), with concentrations decreasing along a
salinity gradient in the estuaries of the Scheldt (Van der Sloot et al., 1985), Test
(Measures and Burton, 1978), Rhone (Guan and Martin, 1991), and James
(Takayanagi and Wong, 1984) rivers. DISe has also been shown to behave
non-conservatively, with concentrations decreasing along the salinity gradient
in the Changjiang estuary (Chang et al., 2016), Zhujiang estuary (Yao et al.,
2006), Mex Bay (Abdel-Moati, 1998), San Francisco Bay (Cutter, 1989a), and
Kaoping and Erhjen estuaries (Hung and Shy, 1995). The marine endmember
of the DISe concentrations in the sampled estuaries (salinity > 31) was 0.30
nmol $L^{-1}$ (range: 0.12 to 0.47 nmol $L^{-1}$), encompassing or close to the values
reported for surface water in the South China Sea (around 0.38 nmol $L^{-1}$,
Nakaguchi et al., 2004) and the Pacific (mean of 0.24 nmol $L^{-1}$, range: 0.02 to
0.69 nmol $L^{-1}$) (Cutter and Bruland, 1984; Sherrard et al., 2004; Mason et al.,
2018). The salinity-related increases in DISe in a seaward direction indicate
that the patterns of distribution of DISe in the Rajang, Maludam, Sebuyau, and
Samunsam estuaries are controlled mainly by conservative mixing of ocean-
derived DISe. Nitrate behaves in a similar way in the Dumai River estuary
(Sumatra, Indonesia), another tropical blackwater river (Alkhaitb and
Jennerjahn, 2007). The Maludam, Sebuyau, Samunsam, and Simunjan rivers
are peat-draining rivers, and most of the coastal areas in the Rajang delta are



also covered by peat; thus, DISe in the black-water estuaries and in the Rajang
may have been mainly ocean derived.

In the Rajang estuary, DISe was removed in March and was added in

September during estuarine mixing (Fig. 3g and j). DISe removal rates of
between 52% and 74% were calculated for Maludam River water in the
laboratory mixing experiments (Fig. 6a). At low salinity, DISe concentrations
were slightly scattered around the linear mixing model for the Maludam and
Samunsam rivers (Fig. 5c and 4c). The Rajang River drains mineral soils in its
upper reaches, and peatland is found only in the delta surrounding the
distributaries (Staub et al., 1994, 2000). In the distributary channels, DOC
concentrations reached around 240 $\mu$mol L$^{-1}$, almost double the concentrations
further upstream, indicating large inputs of organic matter from peat. High DOC
concentrations (between 3100 and 4400 $\mu$mol L$^{-1}$) have also been reported for
the Maludam River (Martin et al., 2018). As discussed above, organic matter
can immobilize Se(IV) and Se(VI). Laboratory studies have shown that Se(IV)
can be adsorbed by peat and that 60% of the adsorbed Se(IV) can be desorbed
upon exposure of the solid phase to seawater (Kharkar et al., 1968). Selenium
may have been added to the Rajang estuary in September via release of Se(IV)
from peat in brackish waters. Other studies have reported removal of the humic
fractions of DOM, colloidal iron, and phosphorus by flocculation in the river–sea
mixing zones (Eckert and Sholkovitz, 1976; Forsgren et al., 1996; Asmala et al.,
2014). Some of the DISe may exist in colloidal form in natural water
(Takayanagi and Wong, 1984), and DISe may be removed by flocculation. In
peat-draining estuaries, ocean-derived DISe may be adsorbed to peat and may
be associated with DOM, which is then converted to DOSe and/or flocculated
to particulate Se.





### 4.3 Behaviour of DOSe during estuarine mixing


In contrast to DISe, DOSe concentrations were highest in the rivers and
decreased in a seaward direction as salinity increased, indicating a terrestrial
origin of DOSe. During estuarine mixing in other estuaries, DOSe has been
shown to behave non-conservatively, with concentrations decreasing along
salinity gradients in the Rhone estuary (Guan and Martin, 1991), Mex Bay
(Abdel-Moati, 1998), and Kaoping and Erhjen estuaries (Hung and Shy, 1995),
and with mid-estuarine input in the San Francisco Bay (Cutter, 1989a). DOSe
concentrations in the estuaries studied in Sarawak were higher than those
reported in other estuaries, such as the Rhone, Kaoping, and Erhjen estuaries
(0.1 to 0.7 nmol L$^{-1}$) (Guan and Martin, 1991; Hung and Shy, 1995), and in San
Francisco and Mex bays (0.1 to 2.5 nmol L$^{-1}$) (Cutter, 1989a; Abdel-Moati,

1998).

4.3.1 Rajang estuary
In the Rajang estuary, DOSe exhibited non-conservative mixing, and
DOSe concentrations in most of the brackish waters were higher than the TML
values (Fig. 3g, 3j). Compared with the TML, removals of DISe were greater
than additions of DOSe in the distributary channels, indicating that not all of the
DOSe was from the conversion of DISe. High DOSe concentrations observed
in coastal areas such as San Francisco Bay have been attributed to *in situ*
production of DOSe by phytoplankton (Cutter, 1989a). However, chlorophyll-a
concentrations in our study area very rarely exceed 2.5 µg L$^{-1}$ and are
ubiquitously 1 µg L$^{-1}$ in the Rajang (Martin et al., 2018), which means that
phytoplankton production is not a major source of DOSe in Sarawak.
In the Rajang delta, DOC also exhibited non-conservative mixing, with
additions from peatlands in the delta areas (Martin et al., 2018). Leaching from





peat soils in the delta areas may be an important source of DOSe in estuarine
mixing zones of the distributary channels in the Rajang estuary. However, there
was no significant correlation between DOSe concentration and the CDOM
spectral slope from 275 to 295 nm ($S_{275-295}$, data from Martin et al., 2018),
specific UV absorbance at 254 nm ($SUVA_{254}$, data from Martin et al., 2018), or
the humic-like C3 component (data from Zhou et al., 2019) in the Rajang
estuary for March ($p > 0.05$) (Fig. 8a-c). The maxima excitation wavelength of
the humic-like component C1 was 330 nm, which is higher than the maxima
excitation wavelength of the humic-like component C2 (275 nm) (data from
Zhou et al., 2019), showing that the C2/C1 ratio is enhanced by
photodegradation (Wang et al., 2019). DOSe/DOC and DOSe/DISe ratios were
not correlated with C2/C1 ratios ($p > 0.05$) (Fig. 8d and e). As suggested by
Martin et al. (2018), sediment loads are high and attenuate very strongly within
the Rajang delta, so the selective removal of a high-molecular-weight CDOM
fraction may be due to sediment adsorption rather than photodegradation.
DOSe was added to, rather than removed from, the brackish waters in the
Rajang estuary (Fig. 3c and f).
4.3.2 Peat-draining rivers and estuaries

The blackwaters in Sarawak are characterized by high levels of terrigenous

DOM with high average molecular weight and high aromaticity (Martin et al.,
2018). In the peat-draining Maludam River, DOSe concentrations were
negatively correlated with $S_{275-295}$ (data from Martin et al., 2018) and $SUVA_{254}$
(data from Martin et al., 2018) during estuarine mixing in both seasons (Fig. 8f
and 8g), which differed from the pattern observed for the Rajang. DOC
concentrations ranged from 96 to 200 $\mu$mol $L^{-1}$ in the Rajang estuary and were
nearly 20 times lower than those in the peat-draining river estuary (the Maludam,
range: 256 to 4386 $\mu$mol $L^{-1}$) in March, and the CDOM concentration and C1–



C5 components in the Rajang estuary were also almost 10 times lower than
those in the Maludam estuary (Martin et al., 2018; Zhou et al., 2019), indicating
that, depending on the geochemical setting, the concentration and molecular
weight distribution of DOM in the Rajang estuary were unlike those in the
Maludam. $S_{275-295}$ is inversely related to the mean molecular weight of DOM
(Helms et al., 2008), and $SUVA_{254}$ is positively related to the aromaticity of DOM
(Weishaar et al., 2003). These correlations suggest that DOSe was associated
closely with high-molecular-weight and highly aromatic DOM in the Maludam
estuary. Also, DOSe concentrations were strongly and positively correlated with
the humic-like C3 component (data from Zhou et al., 2019) during estuarine
mixing in the two seasons (Fig. 8h). As reported by Zhou et al. (2019), the C3
components derived corresponded to aromatic and black carbon compounds
with high molecular weight. The positive correlation between DOSe and the C3
component (Fig. 8g) also indicates that DOSe fractions are associated with
high-molecular-weight aromatic DOM in the Maludam estuary (Fig. 9). Kamei-
Ishikawa et al. (2008) investigated the binding between Se(IV) and humic acid
in a laboratory study and found that the Se remaining in solution was associated
with the dissolved humic acid fractions, and those authors' ultrafiltration
experiments suggested that 50% to 60% of these Se–humic acid associates
had high molecular weights (>10 kDa). Bruggeman et al. (2007) studied the
interactions between Se(IV) and humic substances in aqueous sediment
extracts and found that consistent with our findings, over 70% of the original
Se(IV) was associated with high molecular weights (>30 kDa). Gustafsson and
Johnsson (1994) reported that forest soil fixed 10% of added Se(IV) into low-
molecular-weight fractions of the humic substances by means of microbial
reductive preferential incorporation.

In the Maludam estuary, DOSe/DOC ratios were negatively correlated with

C2/C1 ratios (Fig. 8j), indicating that compared to bulk DOM, the DOSe



fractions were more susceptible to photodegradation during estuarine mixing;
similarly, other researchers have found that aromatic DOM structures are
particularly photoreactive (Ospal and Benner, 1998; Stubbins et al., 2012). As
shown in Fig. 8k, DOSe/DISe ratios are also correlated negatively with C2/C1
ratios, indicating that DOSe was probably photodegraded to DISe. Thus,
photodegradation plays an important role in DOSe processing in the study area,
and DOSe might contain a significant photoreactive fraction that facilitates
photodegradation of DOSe into lower mean molecular weights or gaseous Se
or photomineralization to DISe (Fig. 9). Considerable amounts of Se may be
volatilized when methylselenide compounds form (Lin et al., 2003; Lidman et
al., 2011). A field study conducted in Switzerland found that volatile species of
Se, including dimethyl selenide, dimethyl diselenide, methane seleninic acid,
and dimethyl selenoxide, were naturally emitted from peatland at
concentrations of around 33 nmol $L^{-1}$ (Vriens et al., 2015). As a result of the
method used in the present study, volatile methylselenide compounds in the
DOSe fractions may not have been detected, so DOSe may have been
underestimated. In future work, particular attention should be given to
methylselenide. Martin et al. (2018) suggested that because of the short
residence time in rivers, most photodegradation of terrestrial DOM in the rivers
of Sarawak likely took place after it reached the sea rather than within the rivers
and estuaries. Studies have shown that photodegradation of DOM results in a
range of bioavailable products (Miller and Moran, 1997). In the coastal areas of
the Sarawak, the high temperature promotes rapid microbial metabolism, the
residence time is longer, and solar irradiation is high (Martin et al., 2018). Once
transported to offshore, peatland-derived DOSe might be degraded to a lower
molecular weight or DISe, both of which are bioavailable for phytoplankton and
may enhance the productivity of oligotrophic waters.

The marine endmember of the DOSe concentrations in the sampled



estuaries (salinity >31) ranged from 0.42 to 2.91 nmol $L^{-1}$ (mean: 1.32 nmol $L^{-1}$)
and exceeded those in surface water of the South China Sea (mean: 0.20 nmol
$L^{-1}$, range: 0.33 to 0.14 nmol $L^{-1}$, Nakaguchi et al., 2004) and the Pacific (mean:
0.36 nmol $L^{-1}$, range: 0.01 to 0.67 nmol $L^{-1}$ (Cutter and Bruland, 1984; Sherrard
et al., 2004; Mason et al., 2018). The high DOSe concentrations in coastal
waters in Sarawak (S > 30) suggest a significant contribution from terrigenous
DOSe. Several studies have observed that phytoplankton can excrete organic
selenides after assimilating Se(IV) (Vandermeulen and Foda, 1988; Besser et
al. 1994; Hu et al. 1997). As described by Cutter and Cutter (1995, 2001), DOSe
in ocean surface waters, which is associated mainly with soluble peptides,
appears to have a low molecular weight and, as shown in a laboratory study
(Baines et al., 2001), can be taken up again by phytoplankton at rates of
between 4% and 53%. Photoreactive DOSe fractions, in which Se may be
associated with aromatic and black carbon compounds with high molecular
weight, are discharged offshore by peat-draining rivers (Fig. 8g), from where
they are probably transported across the marginal sea and circulated globally
(Fig. 9). Given that the bioavailability and biogeochemical cycling of the
peatland-derived DOSe fractions may differ from those of peptides produced *in*
*situ* by phytoplankton in the ocean, the impact on coastal and open ocean
ecosystems should be evaluated in the future.

## 665    5. Conclusion

To the best of our knowledge, this is the first study of seasonal variations
in Se speciation in peat-draining rivers and estuaries in Southeast Asia.
Contrary to our expectations and the results from studies elsewhere, DOSe, not
DISe, was the major species in the peat-draining rivers and estuaries of
Sarawak, Malaysia. In blackwater estuaries, DISe was positively related to
salinity, indicating a marine origin, and DOSe was negatively related to salinity,



indicating terrestrial sources. In the delta area of the Rajang River, where
peatland dominates, DISe concentrations increased with salinity, and DOSe
concentrations generally decreased with salinity but increased in the middle
parts of the estuary. In the Maludam, DOSe fractions may be associated with
high-molecular-weight peatland-derived aromatic and black carbon compounds
and may photodegrade to more bioavailable forms once transported to
oligotrophic coastal waters, where they may promote productivity.

## 680    6. Author contribution

JZ, MM, YW, SJ and YC conceptualized the research project and planned
the field expeditions. SJ, AM, EA, FJ and MM performed sample collection and
in-situ measurement for the cruises. YC, WWC, JGQ, JLR, EMR and XLW
completed laboratory analyses. YC, XNW, YW, JS, JZ and MM processed and
analysed the data. All co-authors participated in the interpretation and
discussion of the results. YC prepared the manuscript with suggestions from all
co-authors

## 688    7. Competing interests

The authors declare that there is no conflict of interesting.

## 690    8. Acknowledgements

This study was kindly supported by the Newton-Ungku Omar Fund
(NE/P020283/1), Natural Science Foundation of China (41806096), China
Postdoctoral Science Foundation (2018M630416; 2018M632062), MOHE
FRGS 15 Grant (FRGS/1/2015/WAB08/SWIN/02/1), Overseas Expertise
Introduction Project for Discipline Innovation (B08022), SKLEC Open Research



Fund (SKLEC-KF201610) and Scientific Research Foundation of SKLEC
(2017RCDW07). The authors thank the Sarawak Forestry Department and the
Sarawak Biodiversity Centre for permission to conduct collaborative research
in Sarawak waters under permit numbers NPW.907.4.4(Jld.14)-161, Park
Permit No WL83/2017, and SBC-RA-0097-MM. Lukas Chin and the
SeaWonder crew are acknowledged for their support during the cruises.
Technical support from Dr Patrick Martin and Dr Gonzalo Carrasco at Nanyang
Technological University during the cruises is gratefully acknowledged.

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





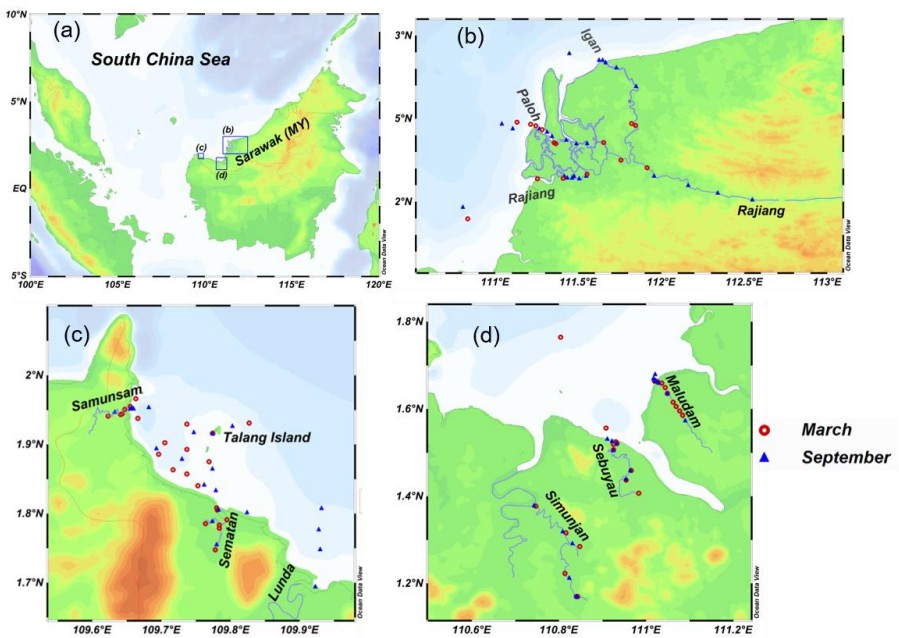


Figure 1. (a) Map of the study area showing the location of Sarawak on the
island of Borneo. Blue boxes with letters indicate the areas shown in panels
b–d. (b–d) Station locations for the Rajang River (b), the Samunsam,
Sematan, and Lunda rivers (c), and the Maludam, Sebuyau, and Simunjan
rivers (d) in March and September 2017. The maps were made with Ocean
Data View (2019).



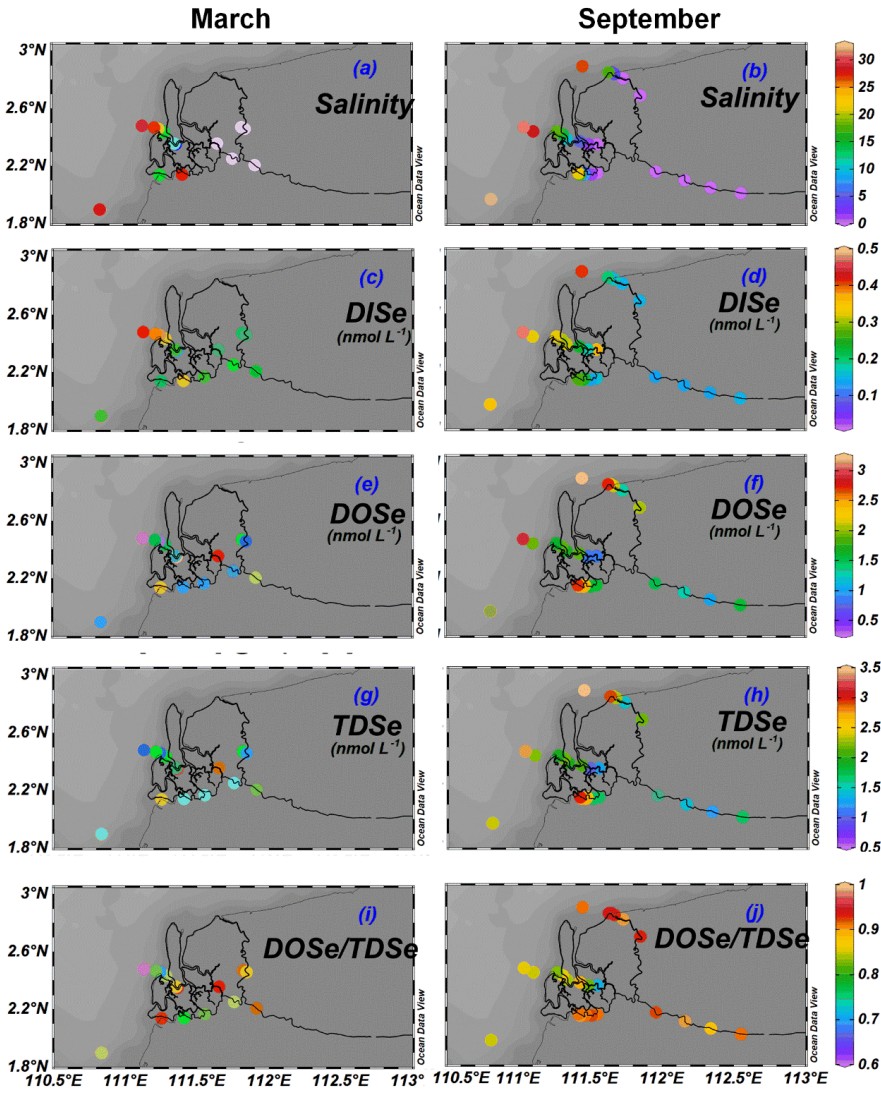


Figure 2. Distributions of salinity, of DISe, DOSe, and TDSe concentrations,

and of DOSe/TDSe ratio in surface waters of the Rajang estuary in March and

September 2017. All distribution plots were made with Ocean Data View

(2019).

969



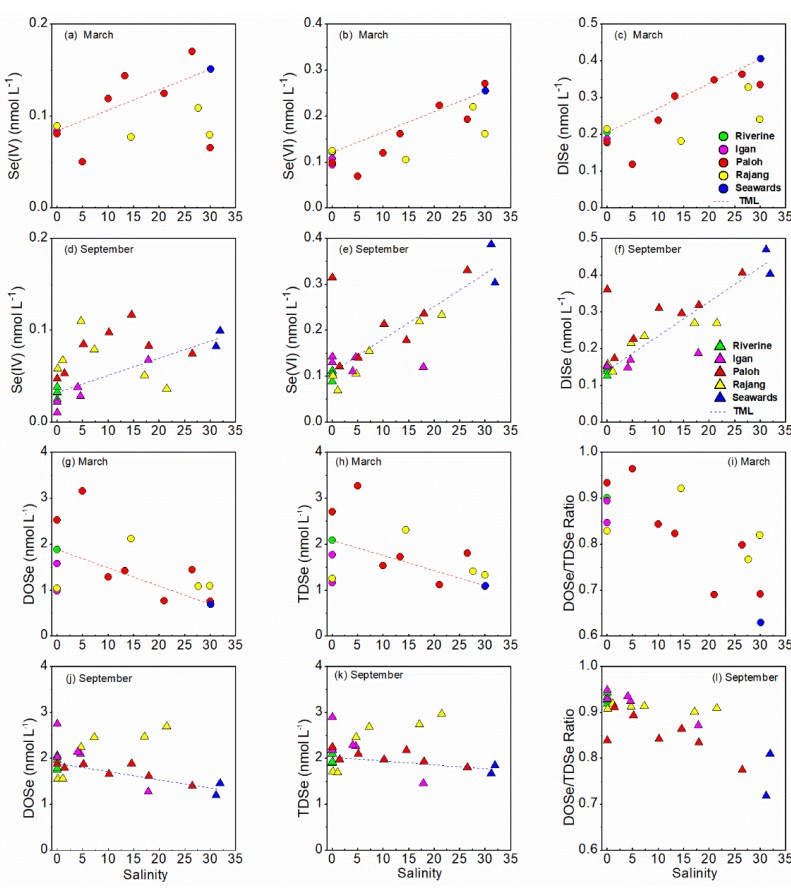

Figure 3. Relationships between Se(IV) (a, d), Se(VI) (b, e), DISe (c, f), DOSe

(g, j), and TDSe (h, k) concentrations, and DOSe/TDSe ratio (i, l) with salinity

in the Rajang and three Rajang tributaries (Igan, Lassa, and Rajang) in March

and September 2017. TML refers to the theoretical mixing line, which was

defined using two endmembers: freshwater in the riverine system and

seawater.




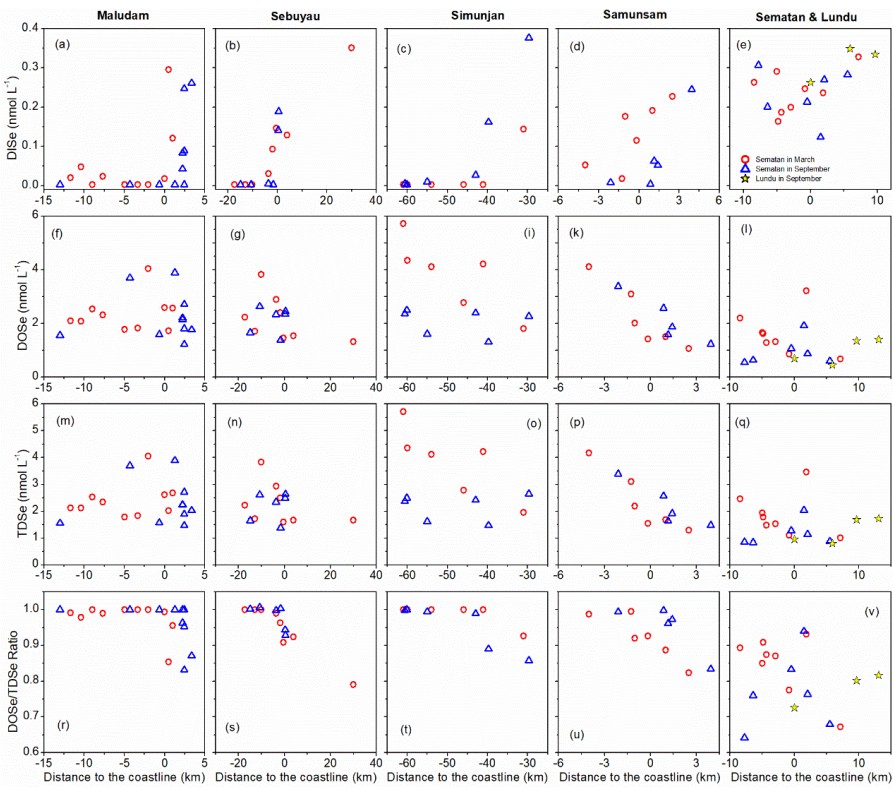


Figure 4. DISe, DOSe, and TDSe concentrations and DOSe/TDSe ratio along
the Maludam, Sebuyau, Simunjan, Samunsam, Sematan, and Lunda
estuaries in March and September 2017.




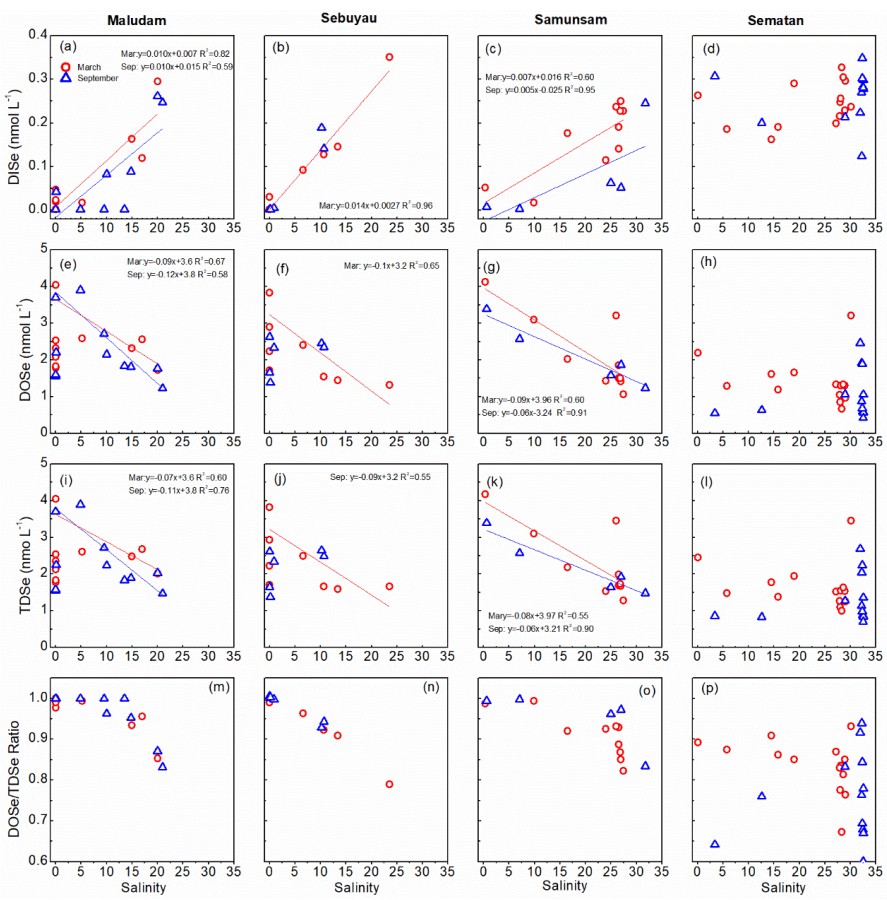



Figure 5. Relationships between DISe, DOSe, and TDSe concentrations and
DOSe/TDSe ratio with salinity in the Maludam, Sebuyau, Samunsam, and
Sematan estuaries. Red circles and blue triangles represent data for March
and September 2017, respectively.






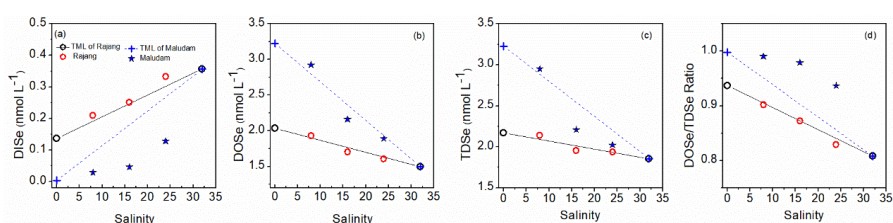


Figure 6. Results of laboratory mixing experiments showing variation in DISe,

DOSe, and TDSe concentrations and DOSe/TDSe ratio as a function of

salinity using filtered riverine water from the Rajang and Maludam rivers and

filtered coastal seawater. TML refers to theoretical mixing line.

998


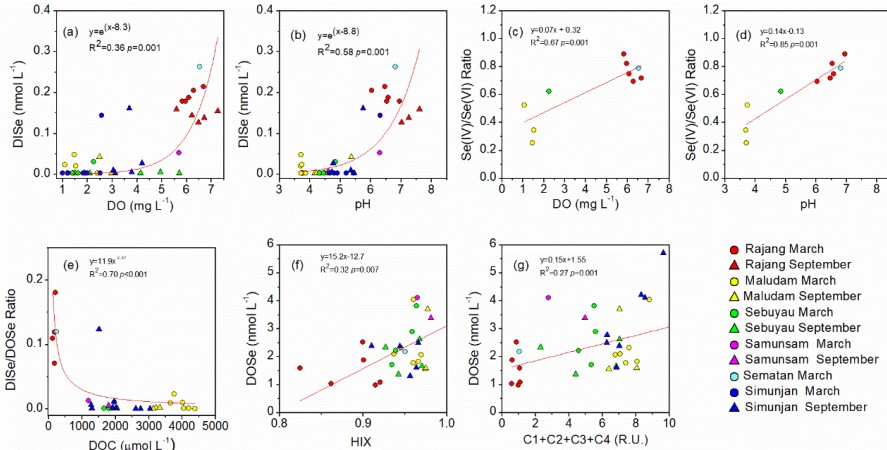

999

Figure 7. Relationships between (a, b) DISe concentrations and DO and pH values, (c, d) Se(IV)/Se(VI) ratios and DO and pH values, and (e–g) DOSe concentrations with the humification index (HIX) and the sum of humic-like CDOM components (C1, C2, C3, and C4) in freshwater (Salinity < 1) for the Rajang, Sematan, Maludam, Sebuyau, Samunsam, and Simunjan rivers in March and September. The HIX and C1, C2, C3, and C4 components are from Zhou et al. (2019) from the same cruises. DO concentrations and pH values were not available for the Sematan River for September, and the HIX and CDOM components were not available for the Rajang River for September. Se(IV)/Se(VI) ratios were calculated only if Se(IV) and Se(VI) concentrations were both above the detection limits, meaning that the data are limited.






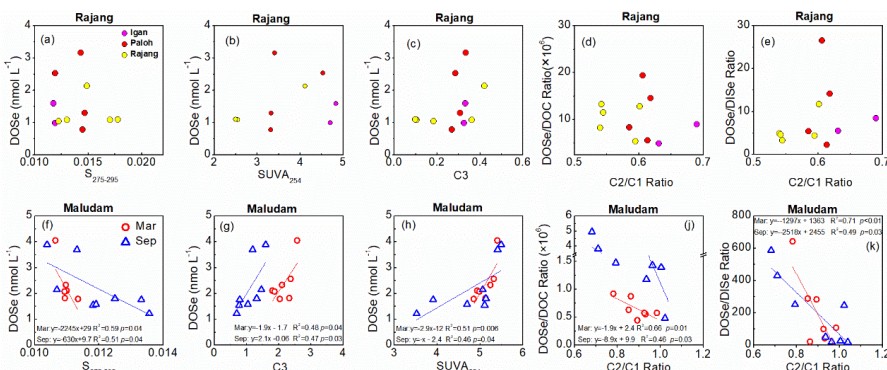


Figure 8. Relationships between DOSe concentrations and $S_{275-295}$, C3
components and $SUVA_{254}$, DOSe/DOC ratio and C2/C1 component ratios,
and DOSe/DISe ratios and C2/C1 component ratios in the Rajang and
Maludam estuaries. The $S_{275-295}$, $SUVA_{254}$, C1, C2, and C3 components are
from Martin et al. (2018) and Zhou et al. (2019) from the same cruises.

Biogeosciences Open Access
Discussions

EGU

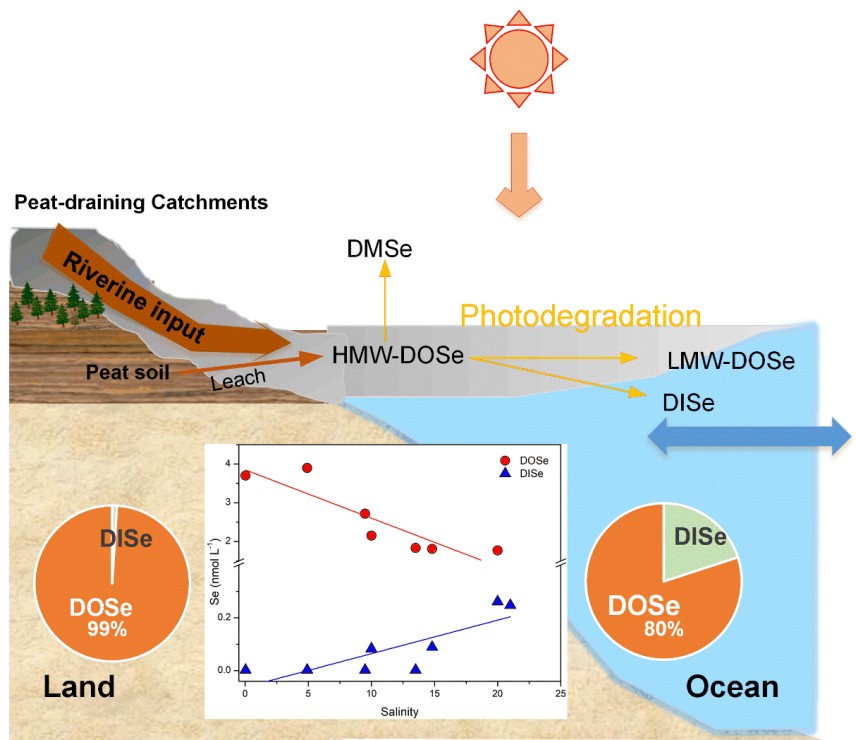

Figure 9. Conceptual diagram of the behaviour of Se species in the Maludam
estuary. HMW, LMW, and DMSe represent high molecular weight, low
molecular weight, and dimethyl selenide, respectively.
