# Peer review of "Distribution and behaviour of dissolved selenium in"

_Biogeosciences, 2019_

## Referee Comment (RC1) · O.S. Pokrovsky (Referee) · 6 Aug 2019

The manuscript of Chang et al presents a combination of field study and laboratory experiments. This work reports first analysis of the distribution and behavior of dissolved Se in several rivers and estuaries of western Borneo. The concentration and speciation of Se were assessed using state of the art analytical techniques. The authors also evaluated the fate of Se species in estuaries and characterized the DOSe in rivers and estuaries. Synthetic figure 9 is especially appreciated. The origins of DISe and DOSe seem to be correctly identified. Specific comments to consider for improvement of ms: L74-87: This is too detailed literature review, not directly linked to the subject of this

study. It is probably not necessary. Physio-geographical parameters of rivers should be listed in a table (% of coverage by peat, degree of affection by palm plantations, runoff, slope etc) Three type of Se behavior are well identified and summarized in L 349-358. However, the presentation of each individual river in Figs 2-5 takes too much space. Either consider presenting just an example of each group or the average of all rivers in each group L407-412: Please explain, what is the mechanism of Se(IV)/Se(VI) increase with DO increase. Oxidation is more pronounced at high DO, yet the observations are reverse to that. L434-439: This information should be in the site description table of river watershed parameters L468-477: There are certainly some structural data (e.g., XAS) on molecular status of Se bound to organic matter. L505-507: The analogy with NO3 is not straightforward: nitrate is a nutrient but Se is not always a nutrient Fig 8a-c present the data from other papers and as such not necessary. Citations of main results from these papers would be enough. L580-643: Basically, the same comment. Too many specific details from other papers; the whole section can be greatly shortened, and only main findings are presented. L621-625. This conclusion is true, however it is based on very indirect observations (many parameters are from already published works). Note that the main source of Se in peatland waters as from highly aromatic DOM of peat horizons has been recently evidenced in Siberian lakes (Pokrovsky et al., 2018 Env Sci Technol) Fig 2: How representative is Rajang to other rivers, why it is shown? Fig 3 is fine Fig 4 might not be needed - may be in Supplement? Previous Fig 3 is way more informative. Fig.4 should be shortened, at least. Fig. 6: what is the difference with fig 3? (hard to apprehend) Fig 6: The size of panels is too small, please enlarge Fig. 8: The plots showing no relationships between variables are not needed to be shown; it is enough just to state that there is no link between variables. General comment: The authors could present the fluxes of Se to the ocean, in different forms. The yield from watersheds of different rivers (i.e., in kg/km$^2$/y) could be compared with that of other large and small rivers of the world, if the data are available. How important are small rivers of Borneo on a global scale of DISe and Dose delivery to the ocean? Are the yields disproportionally high? Conclusions nicely reflect

Interactive
comment

the main findings, and even if some of them are speculative (L 675-678), they can be stated as they are. Oleg S Pokrovsky

---

## Referee Comment (RC2) · Anonymous Referee #2 · 16 Aug 2019

In this manuscript, the authors report Se partitioning into dissolved Se(IV), Se(VI), total inorganic and organic fractions in seven rivers and their estuaries in Sarawak, Malaysia. Furthermore, they conducted a mixing experiment of river and sea water to determine the change in dissolved Se speciation. Overall, I think that the paper adds novel information to our knowledge of the Se cycle. However, it has largely a descriptive character, which could be changed by coming up with a number of hypotheses that can be tested with the data. Moreover, the original results are shown in too much detail, which obscures the general findings. I suggest to come up with figures that combine the results from several or all studied locations and move the original data to the supplementary information.

As already indicated in my preliminary review, the paper suffers from a number (of minor) technical problems. It should be strictly structured according to the objectives, which is not the case in the introduction where the state-of-the-art concerning Objective 3 is not introduced. It is also not the case in the discussion and conclusion sections. The discussion sections includes results referring to Figs. 7 and 8, which need to be moved to the results section. Finally, the manuscript should be shortened, e.g., by moving all content that is mainly of local interest to the supplementary information.

In addition to these general remarks, I offer a number of line-by-line comments: l. 25 and l. 52: Please be clear about which organisms really need Se. I know that mammals and humans need it. At the same time plants do not need Se. I am not familiar with marine organisms. Please specify, which marine organisms need Se. I would not have expected that Se is essential for phytoplankton (because it is not for plants). This question is important and should be clearly addressed. l. 37: What do you mean by "extremely"? Add numbers. l. 47: I am not sure if the introduction of Se can generally promote productivity. This would only be possible, if Se was an essential nutrient for the considered organism. Furthermore, growth would only be promoted if Se was the limiting element, but other limitations are more likely (e.g., by P or Fe). l. 48-49: I don't understand this conclusion. I would prefer a conclusions, which is derived from your results. l. 54-67: About which organisms are you talking? This cannot be generalized! l. 57-59: Please explain this hypothesis. Its understanding is related with my above criticsm, that there is no detailed explanation for which organisms Se is necessary and for which not. l. 64: Why is "organic selenide" mentioned separately? It is included in the oxidation state –II. l. 68: Does phytoplankton really need Se? l. 88: What do you mean by "various", the previously cited studies? Perhaps better cite them again. l. 94: Do you mean that "Se speciation" was controlled? l. 97: "formation" or "generation" instead of "regeneration" l. 106: How does organic matter influence the bioavailability and fate of Se? l. 110: What do you mean by "behaviour"? l. 122: The third objective "falls from heaven". l. 176: Why did you remove the colloids from the seawater samples? Doesn't this result in a rather artificial experiment in which some chemical transformations that

can occur in the environment are ruled out? Please explain. Furthermore, I suggest to come up with a hypothesis, e.g., pure mixing vs. chemical transformations (which?). l. 210: Did you check for normal distribution and transform the data if necessary? l. 214-235: I am a bit lost here. Perhaps, this can be concentrated to the information in aggregated form that is really necessary to understand the results, thereby shortening it. l. 223 and 231: Shouldn't the numbers of the supplementary figures be switched (according to the sequence of their reference in the manuscript)? l. 250-268: I suggest to show the results as bar diagram with error and indication of statistically significant difference instead of the current Fig. 2, which I suggest to move to the supplementary information. l. 349-358: I suggest to combine the results of each of your three groups into a figure for the group instead of showing all individual results. l. 384ff: The Discussion section should be structured according to the three objectives into three parts. The objectives should be discussed as concisely as possible, i.e. the current discussion should be shortened. l. 407-412: This belongs to the results. l. 432-433: Why is this information important? I suggest to delete it. l. 434-441: This is a repetition. Delete and focus on the important statement in l. 441-442. l. 469: You can omit "as follows" and directly start with the numbered list. l. 479: Combine 4.1 and 4.2 as joint contribution to Objective 1. l. 541/545: Isn't the Rhone forming a delta? Or don't you talk about the French Rhone? l. 549: Skip this heading to avoid overstructuring. l. 562: I am confused by the simultaneous use of delta and estuary, because I think that these are two contrasting geomorphological forms, mainly driven by the strength of the tide. l. 563-573: I would move large parts of this and the associated figure to the results section. l. 588-589: The numbers should be subscripts. l. 593-601: I would again move large parts of this and the associated figure to the results section. l. 622: Will photodegradation really be important in the dark DOM-rich waters? Possibly, it is restricted to the uppermost surface-near few mm. l. 665-678: The conclusions should fit to the objectives, i.e. there should be three main conclusions and perhaps a kind of outlook. l. 678: See my previous comments to the role of Se for biological productivity. Figs. 3-5 show all individual results. I suggest to aggregate these data in a way that

clearly illustrates your main points. Figs. 7 and 8 should be included in the Results section.

---

## Author Comment (AC2) · 19 Oct 2019

| 1  | Response to review                                                              |
|----|---------------------------------------------------------------------------------|
| 2  |                                                                                 |
| 3  | Respect Anonymous Referee #2                                                    |
| 4  | We want to begin by thanking Referee #2 for writing that "I think that the      |
| 5  | paper adds novel information to our knowledge of the Se cycle." We are          |
| 6  | extremely grateful for his/her insightful advice and elaborate revisions of the |
| 7  | manuscript. We addressed all the points raised by the referee, as summarized    |
| 8  | below.                                                                          |
| 9  |                                                                                 |
| 10 | General Comments:                                                               |
| 11 | Overall, I think that the paper adds novel information to our knowledge of the  |
| 12 | Se cycle. However, it has largely a descriptive character, which could be       |
| 13 | changed by coming up with a number of hypotheses that can be tested with        |
| 14 | the data. Moreover, the original results are shown in too much detail, which    |
| 15 | obscures the general findings. I suggest to come up with figures that combine   |
| 16 | the results from several or all studied locations and move the original data to |
| 17 | the supplementary information. As already indicated in my preliminary review,   |
| 18 | the paper suffers from a number (of minor) technical problems. It should be     |
| 19 | strictly structured according to the objectives, which is not the case in the   |
| 20 | introduction where the state-of-the-art concerning Objective 3 is not           |
| 21 | introduced. It is also not the case in the discussion and conclusion sections.  |
| 22 | The discussion sections includes results referring to Figs. 7 and 8, which need |
| 23 | to be moved to the results section. Finally, the manuscript should be           |
| 24 | shortened, e.g., by moving all content that is mainly of local interest to the  |
| 25 | supplementary information.                                                      |
| 26 | Response: Thanks for the great advice. We have carefully revised the            |
| 27 | manuscripts based on the comments.                                              |

We have come up three hypotheses, 1) the DOSe is the major species in 29 those peatland- draining rivers; 2) the source of DOSe probably is peat soils 30 and 3) Large amounts of TDSe from peatland-draining rivers were delivered 31 to the coastal water. Those hypotheses were tested with data, as shown in 32 results 3.2 and discussion 4.1, DOSe/TDSe ratios ranged from 0.56 to 0.99, 33 indicating that DOSe was the major species of Se in the peat-draining rivers 34 and estuaries (Fig. 2). The relationship between DOSe and HIX, humic-like 35 CDOM components, S275-295 and SUVA254 (Fig. 4, Fig 5) indicating that peat 36 soils is inferred to be the major source of DOSe in our sampled rivers and 37 DOSe may be associated with with high-molecular-weight and highly humic 38 substances. These was demonstrated in results 3.4 and discussion 4.2. The 39 TDSe flux was estimated in the discussion 4.3. The results showed that TDSe 40 delivered from Rajang were less than those large rivers, but exceeded other 41 small rivers reported so far (Table 2). As for DOSe yields for Rajang and 42 Maludam were one or even two orders of magnitude higher than other 43 reported rivers so far (Table 2). This indicates that the numerous small 44 blackwater rivers draining from peatland are very efficient TDSe and DOSe 45 sources for the coastal waters.

We have deleted some details of the results and simplified the manuscripts. The result 3.1 to 3.3 were down sized from 145 to 84 lines for now, and the figures were downsized from nine to six by moving two original figures (Fig. 2 and 4) to supplementary, and we also have combed two original figures (Fig. 3 and 5) to one figure by presenting the three typical salinity-concentrations relationships (detailed response was shown in comment 17).

We have revised the objectives and reorganized the discussions
according to the objectives. The three objectives were 1) evaluate the fate of
Se species during estuarine mixing in peatland-draining estuaries; 2)

characterize the DOSe fractions; and 3) estimate the magnitude of Se fluxes 57 delivered from peatland-draining rivers to coastal ocean. The objective 2 (i.e. 58 the original objective 3) is about the character of DOSe, which were added in 59 the introduction (Lines 94 to 104, detailed response was shown in comment 60 11). The discussion sections were structured to three parts according to the 61 objectives, as following: 4.1) Fate of Se species during estuarine mixing; 4.2) 62 Character of the DOSe fractions; and 4.3) TDSe flux. The conclusion also 63 reorganized to fit to the three objectives and an outlook, detail were shown in 64 response 31.

We have moved Figs. 7 and 8 to the results in section "3.4 Correlation
between Se species with DO, pH and DOM", detail was shown in response
19.

The content that is mainly of local interest was moved to the
supplementary information and the original detailed results was deleted. We
have shorted the manuscripts from 1025 lines to 768 lines.

**Comment 1.** I. 25 and I. 52: Please be clear about which organisms really 73 need Se. I know that mammals and humans need it. At the same time plants 74 do not need Se. I am not familiar with marine organisms. Please specify, 75 which marine organisms need Se. I would not have expected that Se is 76 essential for phytoplankton (because it is not for plants). This question is 77 important and should be clearly addressed. 78 **Comment** I. 54-67: About which organisms are you talking? This cannot be 79 generalized! 80 **Comment** 1.57-59: Please explain this hypothesis. Its understanding is related with my above criticsm, that there is no detailed explanation for which organisms Se is necessary and for which not.

**Comment** I. 68: Does phytoplankton really need Se?

*Response:* Similar comments were responded together.

Se is required for biosynthesis of selenocysteine, the twenty first naturally
occurring amino acid in protein (Lobanov et al., 2009). As reviewed by
Lobanov et al (2009), selenoproteins show a mosaic occurrence, with many
organisms, such as vertebrates and algae, having dozens of these proteins,
while other organisms, such as higher plants and fungi, having lost all
selenoproteins during evolution.

Selenium is an essential microelement for all aquatic organisms-92 microorganisms, algae, higher aquatic plants and animals (Bodnar et al, 93 2014). In photosynthetic microorganisms, the essential requirement for 94 selenium has been reported in 33 species belonging to six phyla (Table R1, 95 Araie and Shiraiw, 2016). Price and Harrison (1988) found selenoproteins 96 compounds (GSH-Px) in Thalassiosira pseudonana and confirmed obligate 97 requirement for Se in marine diatom. When Se was added to the culture 98 medium, growth was stimulated in the diatom *Thalassiosira pseudonana*, 99 Chysochromulina breviturrita in Haptophyceae, the dinoflagellates 100 Gymnodinium catenatum and Alexandrium minutum, and other algae (Table 101 R1, Araie and Shiraiw, 2016). Studies showed that diatom (Thalassiosira 102 pseudonana, Chaetoceros) cultures deprived of Se(IV) in seawater for more 103 than 5 days did not recover even when Se(IV) was added afterwards 104 (Harrison et al., 1988). The study concluded that it was more difficult for these 105 Se-dependent microorganisms to recover after exposure to Se depletion than 106 from exposure to nitrogen or phosphorus limitation. Similar results are found 107 in Doblin et al. (1999) where three marine phytoplankton species 108 (Gymnodinium catenatum, Alexandrium minutum, Chaetoceros cf. 109 *tenuissimus*) showed rapid decline in Se deficient seawater, resulting in 110 cessation of cell division after eight weeks of Se(IV) depletion. However, studies on the effect of selenite in the unicellular green alga Chlamydomonas 111

reinhardtii, showed only a little simulative effect on growth (Novoselov et al;

2002). Marine phytoplankton show a stronger trend to a preference for Se than freshwater phytoplankton do (Araie and Shiraiw, 2016).

Geochemical analyses of trace elements in Phanerozoic marine pyrite that sustained periods of severe Se depletion in the past oceans correlate closely with three major mass extinction events, at the end of the Ordovician, Devonian and Triassic periods (Long et al., 2016). Considering the essential of Se for marine phytoplankton, the authors assumed that Se depletion may have been one of several factors in these complex extinction scenarios (Long et al., 2016).

We have changed to "Selenium (Se) is an essential micronutrient for
aquatic organisms", and "marine" have been added before phytoplankton in
the manuscript.

**Table R1.** Phytoplankton species that were demonstrated to require selenium for their

arowtha

|                 | 9.000                                   |
|-----------------|-----------------------------------------|
| Phylum          | Species                                 |
| Diatoms         | Amphiprora hyalina                      |
|                 | Chaetoceros debilis                     |
|                 | Chaetoceros pelagicus                   |
|                 | Chaetoceros vixvisibilis                |
|                 | Coscinodiscus asteromphalus             |
|                 | Corethron criophilum                    |
|                 | Ditylum brightwellii                    |
|                 | Skeletonema costatum (strain 18c NEPCC) |
|                 | Skeletonema costatum (strain 611 NEPCC) |
|                 | Skeletonema costatum (strain 616 NEPCC) |
|                 | Stephanopyxis palmeriana                |
|                 | Thalassiosira pseudonana                |
|                 | Thalassiosira oceanica                  |
|                 | Thalassiosira rotula                    |
|                 | Thalassiosira aestivalis                |
| Dinoflagellates | Alexandrium minnutum b       |
|                 | Gymnodinium catenatum b      |
|                 | Gymnodinium nagasakiense b   |
|                 | Peridinium cinctum fa. Westii           |

|                 | Pyrodinium bahamense b        |
|-----------------|------------------------------------------|
| Prymnesiophytes | Chrysochromulina breviturrita            |
|                 | Chrysochromulina kappa                   |
|                 | Chrysochromulina brevefilum              |
|                 | Chrysochromulina strobilus               |
|                 | Chrysochromulina polylepis b  |
|                 | Helladosphaera sp                        |
|                 | Emiliania huxleyi                        |
|                 | Gephyrocapsa oceanica                    |
| Raphidophytes   | Chattonella verruculosab                 |
| Chlorophytes    | Platymonas subcordiformis                |
| Chrysophytes    | Aureococcus anophagefferens b |

a Modified from Araie and Shiraiw, 2016

b Harmful algae.

**Comment 2** I. 37: What do you mean by "extremely"? Add numbers.

*Response:* We have changed to "the concentrations of DISe were extremely low (near or below the detection limits, i.e. 0.0063 nmol  $L^{-1}$ )".

**Comment 3** I. 47: I am not sure if the introduction of Se can generally promote productivity. This would only be possible, if Se was an essential nutrient for the considered organism. Furthermore, growth would only be promoted if Se was the limiting element, but other limitations are more likely (e.g., by P or Fe).

*Response:* The essential requirement for selenium has been reported in 33

species belonging to six phyla (Table R1, Araie and Shiraiw, 2016). The dominant phytoplankton species was diatom in Sarawak coasts (Saifullah et

- al., 2014). When Se was added to the culture medium, growth was stimulated
- 141 for diatom, and when it cultures deprived of Se(IV) in seawater for more than
- 142 5 days did not recover even when Se(IV) was added afterwards (Harrison et
- al., 1988). Se-limiting for diatom growth were not found in the filed study, although a study in Huon Estuary found that low level Se could be limiting for growth and production of dinoflagellate (Doblin et al., 1999).

We are not sure whether Se could be a limiting element in the Malaysiacoastal waters, thus we have deleted those sentence.

Comment 4 I. 48-49: I don't understand this conclusion. I would prefer a conclusions, which is derived from your results.

**Response:** Thanks. We have deleted the "The results of this study suggest that 152 the impacts of Se discharges on coastal ecosystems should be evaluated in the 153 future", and have changed to "The TDSe flux delivered by the peat-draining 154 rivers exceeded other small rivers, and it is quantitatively more significant than 155 previously thought".

Comment 5. I. 64: Why is "organic selenide" mentioned separately? It is
included in the oxidation state –II.

**Response:** We have deleted the "organic selenide".

**Comment 6** I. 88: What do you mean by "various", the previously cited studies? Perhaps better cite them again.

**Response:** We have cited the previously mentioned studies again.

**Comment 7** I. 94: Do you mean that "Se speciation" was controlled?

*Response:* Yes, we have changed to "Chang et al. (2016) found that Se speciation was controlled by biological, physical, and redox processes in theestuaries".

**Comment 8** I. 97: "formation" or "generation" instead of "regeneration

*Response:* We have changed to "generation of particulate organic selenide in

- 172 the water."
- 173

Comment 9 I. 106: How does organic matter influence the bioavailability and175 fate of Se?

**Response:** We have revised the introduction greatly, and "It is also known that organic matter plays an important role in the bioavailability and fate of Se in the environment" was deleted.

**Comment 10** I. 110: What do you mean by "behaviour"?

*Response:* We have deleted "the behavior", changed to "More studies of Se in fluvial systems in Southeast Asia are therefore needed"

**Comment 11** I. 122: The third objective "falls from heaven".

*Response:* Thanks for the great advices. We have revised the introduction, and added the DOSe research (i.e. objective 2 in the revised manuscript)

status as followed.

"In the high-latitude peatland-draining rivers, dissolved Se concentrations are spatial variable, with concentrations of up to 13 nmol L-1 being observed in 189 northern Minnesota, US (Clausen and Brooks, 1983), from 0.38 to 5 nmol L-1 190 191 in the Krycklan catchment, Sweden (Lidman et al., 2011) and from 0.25 to 1.25 192 in the Siberian (Pokrovsky et al., 2018). Although these various studies did not 193 report different species of Se (Clausen and Brooks, 1983; Lidman et al., 2011; 194 Pokrovsky et al., 2018), the DOSe probably the dominated species in peatland-195 draining river. In the open ocean, DOSe was assumed mainly associate with 196 soluble peptides with low molecule weight in surface waters and were relatively 197 refractory (Cutter and Cutter, 1995; 2004). Substantial amounts of dissolved Se 198 also are known to be associated humic substances, Gustafsson and Johnsson 199 (1994) assumed that Se was preferentially incorporated into low molecular weight humic substances fractions by means of microbial reductive 201 incorporation, while Kamei-Ishikawa et al. (2008) found that Se associated with 202 high molecular weights humic acid fractions. The current paucity of information 203 on DOSe characteristics and its export by rivers from tropical peat-draining 204 rivers remains a major gap in our understanding of Se biogeochemical cycling. 205 Highest concentrations of dissolved organic carbon (DOC) globally were 206 reported in tropical peat-draining rivers in Borneo (Moore et al., 2013; Wit et al., 207 2015). More works of Se in the fluvial systems of this region are therefore 208 needed to provide an improved understanding of the biogeochemical 209 processing of Se and the associations with organic matter."

Comment 12 I. 176: Why did you remove the colloids from the seawater
samples? Doesn't this result in a rather artificial experiment in which some
chemical transformations that can occur in the environment are ruled out?
Please explain. Furthermore, I suggest to come up with a hypothesis, e.g.,
pure mixing vs. chemical transformations (which?).

*Response:* Thanks for the advices. The research aim of this experiment
is to evaluate the impact of particle-free (i.e. dissolved phase) seawater and
river water mixing process on Se species, whether transformation of DISe
between DOSe occurs along salinity gradient. Here, the filter (0.45
micrometer pore size) could retain a significant portion of colloids while only
remove particles.

The results were shown in Fig. R1. The measured DISe, DOSe and TDSe concentrations were comparable with theoretical values, indicating pure mixing in Rajang estuary. However, in Maludam estuary, the measured DISe and TDSe concentrations were lower than the theoretical values, while the measured DOSe concentrations were comparable with the theoretical value. The losses of DISe were not balanced by increasing in DOSe, indicating that chemical transformations between DISe and DOSe did not occur in Maludam.
Other studies have reported removal of the humic fractions of DOM, colloidal
iron, and phosphorus by flocculation in the river–sea mixing zones (Eckert
and Sholkovitz, 1976; Forsgren et al., 1996; Asmala et al., 2014). Some of the
DISe may exist in colloidal form in natural water (Takayanagi and Wong,
1984), and DISe may be removed by flocculation. The removal of DISe were
probably be flocculated to Se particulate form.

This mixing experiment indeed ruled out the impact of particle and part of colloids on the Se transformations. In the future, we would add another set of mixing experiment without filtration, particle-free and particle-included would be designed to the determine the influence of riverine particles and colloids on Se chemical cycling. Thus, considering the incompleteness, this section was deleted here.

Fig. R1. Results of laboratory mixing experiments showing variation in DISe, DOSe, and
TDSe concentrations as a function of salinity using filtered riverine water from the Rajang
and Maludam rivers and filtered coastal seawater. TML refers to theoretical mixing line.

Comment 13. I. 210: Did you check for normal distribution and transform the248 data if necessary?

*Response:* We have checked the normal distribution, and if the data doesn't comply with the normal distribution, Mann Whitney U test were used instead of t-test.

**Comment 14** I.214-235: I am a bit lost here. Perhaps, this can be concentrated to the information in aggregated form that is really necessary tounderstand the results, thereby shortening it.

*Response:* We have shorted the 3.1 section with main findings from 24 lines
to 7 lines. The revised were followed:

"The water chemistry in the freshwater reach of the Maludam, Simunian, and 259 Sebuyau rivers are typical of blackwater rivers draining from peatland with 260 acidic pH and low DO concentrations, and the mixing with coastal water 261 increased the pH and DO (Table S1, Fig. S1). Values of pH and DO 262 concentrations in the Sematan and Lundu, which drain mostly mineral soils, 263 were higher than those in the blackwater rivers (Fig. S1). In the Rajang estuary, 264 values of pH and DO were lower in the riverine side, especially in the 265 distributaries where covered by the peat (Fig. S2)".

**Comment 15** | 223 and 231: Shouldn't the numbers of the supplementary figures be switched (according to the sequence of their reference in the manuscript)?

*Response:* We have switched the numbers of the supplementary figures astheir sequence in the context.

**Comment 16** 250-268: I suggest to show the results as bar diagram with error and indication of statistically significant difference instead of the current Fig. 2, which I suggest to move to the supplementary information.

*Response:* We have moved the current Fig. 2 to the supplementary information, and draw box plot of TDSe, DISe and DOSe concentration and

DOSe/TDSe ratio in the sampled rivers and estuaries (Fig R2, i.e. Fig. 2 in the manuscript).

---

## Author Response (AR1)

**Response to review**

**Respect Dr. Pokrovsky:**

We want to begin by thanking Dr. Pokrovsky for writing that "Conclusions nicely reflect the main findings, and even if some of them are speculative (L 675-678), they can be stated as they are." We greatly appreciated the constructive comments and suggestions to improve the original manuscript. Based on the comments, we have made carefully revision. We addressed all the points raised, as summarized below.

**Comment 1.** L74-87: This is too detailed literature review, not directly linked to the subject of this study. It is probably not necessary.

*Response:* Thanks. We have deleted those sentences in the revised manuscripts.

**Comment 2.** Physio-geographical parameters of rivers should be listed in a table (% of coverage by peat, degree of affection by palm plantations, runoff, slope etc).

*Response:* Thanks for the great advice. We have added those parameters in Table 1. However, the slope of the rivers was not available.

*Revised manuscripts: "*The physio-geographical parameters of sampled river basins are summarized in Table 1" were added in *page 6 line 134-135.* Table 1 were added in *page 25 line 711-725,* as followed.

Table 1 The physio-geographical parameters of sampled rivers. (n.a. stands for not
available.)

| River Names | Total Basin [a] | Runoff $(km^3 yr^{-1})$ | Coverage rate by peat (%) [a] | Degree of affection by palm plantations (%)[a] |
|---|---|---|---|---|
| Rajang | 50000 | 114 [b] | 7.7 | 9.1 |
| Maludam | 197 | 0.14 [c] | 87 | 8.1 |
| Sebuyau | 538 | n.a. | 54 | 4.5 |
| Simunjan | 788 | n.a. | 44 | 30 |
| Samusam | 163 | n.a. | 10 | 0 |
| Sematan | 287 | n.a. | 0 | 0 |

[a] Modified from Bange et al., 2019
[b] Cited from Staub et al., 2000
[c] Cited Müller et al., 2016

**Comment** 3.   Three type of Se behavior are well identified and summarized in L349-358. However, the presentation of each individual river in Figs 2-5

takes too much space. Either consider presenting just an example of each group or the average of all rivers in each group

*Response:* Thanks for the great advice. We have moved Fig. 2 and Fig.

4 to the supplement, and Fig. 3 and Fig. 5 were merged to one figure (Fig. 3

in the revised manuscript). Three typical groups of relationships between Se species and salinity for Rajang, Maludam and Sematan estuaries were selected to present in Fig. 3 in the revised manuscript, and those for Sebuyau and Samunsam were moved to supplement.

*Revised manuscripts:* Figure 3 were present in *page 29 line 738-743,*

as followed.

[Figure]

Fig. 3. Relationships between DISe (a - d), DOSe (e - h), and TDSe (i - l) concentrations
with salinity in the Rajang and three Rajang tributaries (Igan, Lassa, and Rajang), and in
the Maludam and Sematan estuaries in March and September 2017. TML refers to the
theoretical mixing line, which was defined using two endmembers: freshwater in the
riverine system and seawater.

**Comment** 4: L407-412: Please explain, what is the mechanism of

Se(IV)/Se(VI) increase with DO increase. Oxidation is more pronounced at high DO, yet the observations are reverse to that.

*Response:* As shown in Fig. R1, Se(IV)/Se(VI) ratios in the freshwater of the sampled rivers increased as DO concentrations increased statistically.

However, the Se(IV)/Se(VI) ratios were calculated only if Se(IV) and Se(VI)

concentrations were both above the detection limits, meaning that the data are limited. The limited Se(IV)/Se(VI) ratios roughly fell into two groups, one group was low Se(IV)/Se(VI) ratios with low DO concentration and the other was high Se(IV)/Se(VI) ratios with high DO concentration. The liner relationship between Se(IV)/Se(VI) ratios and DO concentration probably was false appearance with the limited data. We have deleted this figure in the manuscript.

[Figure]

Fig. R1 Relationships between Se(IV)/Se(VI) ratios and DO values n freshwater (Salinity

< 1) for the Rajang, Sematan, Maludam, Sebuyau, Samunsam, and Simunjan rivers in

March and September. Se(IV)/Se(VI) ratios were calculated only if Se(IV) and Se(VI)

concentrations were both above the detection limits, meaning that the data are limited.

**Comment** 5: L434-439: This information should be in the site description table of river watershed parameters

*Response:* We have deleted those sentences, table 1 with river watershed parameters were presented as shown in response 1.

**Comment 6**: L468-477: There are certainly some structural data (e.g., XAS)

on molecular status of Se bound to organic matter.

*Response:* Thanks for the advices, we have deleted those sentences, including "However, the mechanisms behind the interactions between Se and dissolved organic ligands are still poorly understood. Three hypotheses have been proposed to explain organic-matter-mediated retention of Se, as follows:

1) direct complexation of organic matter with Se, 2) indirect complexation via

Se-cation–organic-matter complexes, or 3) microbial reduction and incorporation into amino acids, proteins, and natural organic matter (Winkel et al., 2015). Depending on the type of binding, Se may be easily mobilized (e.g., through adjusting pH) or immobilized (e.g., by covalent incorporation to organic matter) (Winkel et al., 2015). However, there is ambiguity about the molecular structure and species of Se that bind to organic matter, and further work is needed to identify the mechanisms by which Se is bound to, and released from, organic matter." in the revised manuscript.

**Comment 7:** L505-507: The analogy with NO3 is not straightforward: nitrate is a nutrient but Se is not always a nutrient

       *Response:* We have deleted the "Nitrate behaves in a similar way in the

Dumai River estuary (Sumatra, Indonesia), another tropical blackwater river (Alkhaitb and Jennerjahn, 2007)."

**Comment 8:**   Fig 8a-c present the data from other papers and as such not necessary. Citations of main results from these papers would be enough.

       *Response:* Consider that the DOSe were not related to the CDOM in the

Rajang, we have deleted the Figure 8a-e, kept Figure 8f-h as Figure 5 in the revised manuscripts.

       *Revised manuscripts:* Figure 5 were present in *page 31 line 757-762,*

as followed.

[Figure]

Figure 5. Relationships between DOSe concentrations and S275-295, C3
components and SUVA254, DOSe/DOC ratio and C2/C1 component ratios,
and DOSe/DISe ratios and C2/C1 component ratios in the Rajang and
Maludam estuaries. The S275-295, SUVA254, C1, C2, and C3 components
are from Martin et al. (2018) and Zhou et al. (2019) from the same cruises.

**Comment 9**: L580-643: Basically, the same comment. Too many specific details from other papers; the whole section can be greatly shortened, and only main findings are presented.

***Response:*** We have deleted those specific literature details and shorted this section from 63 lines to 41 lines.

***Revised manuscripts:*** In *page 15-16 line 373-414,* as followed:

"Moreover the peat-draining rivers demonstrated a liner relationship between

DOSe concentrations and HIX and humic-like CDOM components (Fig. 4d, 4e)

indicating that DOSe may be associated with dissolved humic substances. In addition, DOSe correlated with $S_{275-295}$ and $SUVA_{254}$ (Fig. 5a, 5c) suggesting that DOSe was associated closely with high-molecular-weight and highly aromatic DOM. Also, the positive correlations between DOSe and the humic- like C3 component (Fig. 5b) which derived corresponded to aromatic and black carbon compounds with high molecular weight, also indicates that DOSe fractions are associated with high-molecular-weight aromatic DOM (Fig. 6).

Pokrovsky et al. (2018) also found that Se were transport in the form of high molecular weights organic aromatic-rich complexes from peat to the rivers and lakes in the Arctic. Bruggeman et al. (2007) and Kamei-Ishikawa et al. (2008)

both found that 50% to 70% of Se(IV)–humic substances associates had high molecular weights (>10 kDa), that consistent with our findings.

During the estuarine mixing, the negatively correlation between

DOSe/DOC and DOSe/DISe ratios with C2/C1 ratios which is enhanced by photodegradation (Wang et al., 2019; Fig. 5d, 5e), indicating that compared to bulk DOM, the DOSe fractions were more susceptible to photodegradation, and that DOSe was probably photodegraded to DISe. As suggested by Martin et al (2018) that most photochemical transformations of DOM in Sarawak likely take place after DOM reaches the sea. Thus, photodegradation plays an important role in DOSe processing once transported to offshore, and DOSe might contain a significant photoreactive fraction that facilitates photodegradation of DOSe into lower mean molecular weights or gaseous Se or photomineralization to

DISe (Fig. 6). Considerable amounts of Se may be volatilized when methylselenide compounds form (Lidman et al., 2011). A field study found that volatile species of Se were naturally emitted from peatland at concentrations of around 33 nmol $L^{-1}$ (Vriens et al., 2015). As a result of the method used in the present study, volatile methylselenide compounds in the DOSe fractions may not have been detected, so DOSe may have been underestimated. In future work, particular attention should be given to methylselenide. Studies have shown that photodegradation of DOM results in a range of bioavailable products (Miller and Moran, 1997). Peatland-derived DOSe might be degraded to a lower molecular weight or DISe in the coastal areas, both of which are bioavailable for phytoplankton and may stimulate their growth, and thereby impact the marine animals via food chain. The photoreactive DOSe fractions are probably transported across the marginal sea and circulated globally. Given that the bioavailability and biogeochemical cycling of the peatland-derived DOSe fractions may differ from those of peptides produced *in situ* by phytoplankton in the ocean, the impact on coastal and open ocean ecosystems should be evaluated in the future."

**Comment 10:** L621-625. This conclusion is true, however it is based on very indirect observations (many parameters are from already published works).

Note that the main source of Se in peatland waters as from highly aromatic

DOM of peat horizons has been recently evidenced in Siberian lakes (Pokrovsky et al., 2018 Env Sci Technol)

***Response:*** Thanks for the great advice. We have learned a lot from the literature (Pokrovsky et al., 2018 Env Sci Technol), and also cited in the manuscripts. The investigation of biogeochemical process in the peat-draining rivers and estuaries in Borneo were international cooperation, the CDOM

investigation (Martin et al., 2018; Zhou et al., 2019) cited in our manuscript were conducted by our cooperative partners.

***Revised manuscripts:*** In *page 4 line 84-88,* and *page 15 line 378-387*

as followed:

"In the high-latitude peatland-draining rivers, dissolved Se concentrations are spatial variable, with concentrations of up to 13 nmol $L^{-1}$ being observed in northern Minnesota, US (Clausen and Brooks, 1983), from 0.38 to 5 nmol

$L^{-1}$ in the Krycklan catchment, Sweden (Lidman et al., 2011) and from 0.25 to

1.25 nmol $L^{-1}$ in the Siberian (Pokrovsky et al., 2018)."

"the positive correlations between DOSe and the humic-like C3 component (Fig. 5b) which derived corresponded to aromatic and black carbon compounds with high molecular weight, also indicates that DOSe fractions are associated with high-molecular-weight aromatic DOM (Fig. 6). Pokrovsky et al. (2018) also found that Se were transport in the form of high molecular weights organic aromatic-rich complexes from peat to the rivers and lakes in the Arctic."

**Comment** 11: Fig 2: How representative is Rajang to other rivers, why it is shown?

***Response:*** The Se distribution in the peatland draining estuaries is largely unknown. Compared with other rivers, Rajang is the longest river in

Malaysia, and the delta plain is mainly composed by organic matter enriched sediments which was identified as peat deposits with a maximum depth of 15

m (Staub et al., 2000). Considering the space of the manuscript, Fig. 2 were moved to Supplement.

**Comment** 12: Fig 3 is fine Fig 4 might not be needed - may be in

Supplement? Previous Fig 3 is way more informative. Fig.4 should be shortened, at least.

*Response:* Fig. 4 were moved to supplement; Fig 3 were kept.

**Comment 13:** Fig.6: what is the difference with fig 3? (hard to apprehend) Fig

6: The size of panels is too small, please enlarge

*Response:* Fig.6 is the laboratory mixing experiments that simulated estuarine mixing processes. Fig. 3 is the results of field observations.

Considering the incompleteness of the mixing experiments, Fig.6 was deleted.

**Comment 14:** Fig. 8: The plots showing no relationships between variables are not needed to be shown; it is enough just to state that there is no link between variables.

*Response:* We have deleted the Fig.8 a – e but kept Fig 8. f – k as Fig.5

in the revised manuscripts.

*Revised manuscripts:* In *page 31 line 751,* as followed:

[Figure]

Figure 5. Relationships between DOSe concentrations and $S_{275-295}$, C3 components and SUVA$_{254}$, DOSe/DOC ratio and C2/C1 component ratios, and DOSe/DISe ratios and C2/C1 component ratios in the Rajang and Maludam estuaries. The $S_{275-295}$, SUVA$_{254}$, C1, C2, and C3 components are from Martin et al. (2018) and Zhou et al. (2019) from the same cruises.

**General comment:** The authors could present the fluxes of Se to the ocean, in different forms. The yield from watersheds of different rivers (i.e., in kg/km2/y) could be compared with that of other large and small rivers of the world, if the data are available. How important are small rivers of Borneo on a global scale of DISe and Dose delivery to the ocean? Are the yields disproportionally high? Conclusions nicely reflect the main findings, and even if some of them are speculative (L 675-678), they can be stated as they are.

*Response:* We have added the "4.3 TDSe flux" section in the manuscripts with estimation of the riverine TDSe flux in Table 2 in the manuscript.

[revised manuscript text omitted]

**Response to review**

**Respect Anonymous Referee #2**

We want to begin by thanking Referee #2 for writing that "I think that the paper adds novel information to our knowledge of the Se cycle." We are extremely grateful for his/her insightful advice and elaborate revisions of the manuscript. We addressed all the points raised by the referee, as summarized below.

**General Comments:**

1. Overall, I think that the paper adds novel information to our knowledge of the Se cycle. However, it has largely a descriptive character, which could be changed by coming up with a number of hypotheses that can be tested with the data.

*Response:* Thanks for the great advice. We have carefully revised the manuscripts based on the comments.

We have come up three hypotheses, 1) the DOSe is the major species in those peatland- draining rivers; 2) the source of DOSe probably is peat soils and 3) large amounts of TDSe from peatland-draining rivers were delivered to the coastal water. Those hypotheses were tested with data, as shown in results 3.2 and discussion, DOSe/TDSe ratios ranged from 0.56 to 0.99, indicating that DOSe was the major species of Se in the peat-draining rivers and estuaries (Fig. 2). The relationship between DOSe and HIX, humic-like CDOM components, $S_{275-295}$ and $SUVA_{254}$ (Fig. 4, Fig 5) indicating that peat soils is inferred to be the major source of DOSe in our sampled rivers and DOSe may be associated with with high-molecular-weight and highly humic substances. These was demonstrated in results 3.4 and discussion 4.2. The TDSe flux was estimated in the discussion 4.3. The results showed that TDSe delivered from Rajang were less than those large rivers, but exceeded other small rivers reported so far (Table 2). As for DOSe yields for Rajang and

Maludam were one or even two orders of magnitude higher than other reported rivers so far (Table 2). This indicates that the numerous small blackwater rivers draining from peatland are very efficient TDSe and DOSe sources for the coastal waters.

**Revised manuscripts:** In *page 5 line 108-111,* as followed:

"We hypothesize that the DOSe is the major species in those peatland- draining rivers which mainly from peat soils and sizable Se from peatland is delivered to the coastal areas.

2.   Moreover, the original results are shown in too much detail, which obscures the general findings. I suggest to come up with figures that combine the results from several or all studied locations and move the original data to the supplementary information.

**Response:** We have deleted some details of the results and simplified the manuscripts. The result 3.1 to 3.3 were down sized from 145 to 84 lines for now, and the figures were downsized from nine to six by moving two original figures (Fig. 2 and 4) to supplementary, and we also have combed two original figures (Fig. 3 and 5) to one figure by presenting the three typical salinity-concentrations relationships (detailed response was shown in comment 17).

3.   As already indicated in my preliminary review, the paper suffers from a number (of minor) technical problems. It should be strictly structured according to the objectives, which is not the case in the introduction where the state-of-the-art concerning Objective 3 is not introduced. It is also not the case in the discussion and conclusion sections.

***Response:*** We have revised the objectives and reorganized the discussions according to the objectives. The three objectives were 1) evaluate the fate of Se species during estuarine mixing in peatland-draining estuaries;

2) characterize the DOSe fractions; and 3) estimate the magnitude of Se fluxes delivered from peatland-draining rivers to coastal ocean. The objective

2 (i.e. the original objective 3) is about the character of DOSe, which were added in the introduction (Lines 94 to 104, detailed response was shown in comment 11). The discussion sections were structured to three parts according to the objectives, as following: 4.1) Fate of Se species during estuarine mixing; 4.2) Character of the DOSe fractions; and 4.3) TDSe flux.

The conclusion also reorganized to fit to the three objectives and an outlook, details were shown in response 31.

***Revised manuscripts:*** In *page 5 line 111-114,* as followed:

The main objectives of the study were to 1) evaluate the fate of dissolved

Se species in peatland-draining estuaries; 2) characterize the DOSe fractions; and 3) estimate the magnitude of Se fluxes delivered from to coastal ocean.

4.   The discussion sections includes results referring to Figs. 7 and 8, which need to be moved to the results section.

***Response:*** We have moved Figs. 7 and 8 to the results in section "3.4

Correlation between Se species with DO, pH and DOM", detail was shown in response 19.

5.   Finally, the manuscript should be shortened, e.g., by moving all content that is mainly of local interest to the supplementary information.

***Response:*** The content that is mainly of local interest was moved to the supplementary information and the original detailed results was deleted. We have shorted the manuscripts from 1025 lines to 768 lines.

**Comment 1.** l. 25 and l. 52: Please be clear about which organisms really need Se. I know that mammals and humans need it. At the same time plants do not need Se. I am not familiar with marine organisms. Please specify, which marine organisms need Se. I would not have expected that Se is essential for phytoplankton (because it is not for plants). This question is important and should be clearly addressed.

**Comment** l. 54-67: About which organisms are you talking? This cannot be generalized!

**Comment** l.57-59: Please explain this hypothesis. Its understanding is related with my above criticsm, that there is no detailed explanation for which organisms Se is necessary and for which not.

**Comment** l. 68: Does phytoplankton really need Se?

*Response:* Similar comments were responded together.

Se is required for biosynthesis of selenocysteine, the twenty first naturally occurring amino acid in protein (Lobanov et al., 2009). As reviewed by

Lobanov et al (2009), selenoproteins show a mosaic occurrence, with many organisms, such as vertebrates and algae, having dozens of these proteins, while other organisms, such as higher plants and fungi, having lost all selenoproteins during evolution.

Selenium is an essential microelement for all aquatic organisms- microorganisms, algae, higher aquatic plants and animals (Bodnar et al,

2014). In photosynthetic microorganisms, the essential requirement for selenium has been reported in 33 species belonging to six phyla (Table R1,

Araie and Shiraiw, 2016). Price and Harrison (1988) found selenoproteins compounds (GSH-Px) in *Thalassiosira pseudonana* and confirmed obligate requirement for Se in marine diatom. When Se was added to the culture medium, growth was stimulated in the diatom *Thalassiosira pseudonana*,

*Chysochromulina breviturrita* in Haptophyceae, the dinoflagellates

*Gymnodinium catenatum* and *Alexandrium minutum*, and other algae (Table

R1, Araie and Shiraiw, 2016). Studies showed that diatom (*Thalassiosira*

*pseudonana, Chaetoceros*) cultures deprived of Se(IV) in seawater for more than 5 days did not recover even when Se(IV) was added afterwards (Harrison et al., 1988). The study concluded that it was more difficult for these

Se-dependent microorganisms to recover after exposure to Se depletion than from exposure to nitrogen or phosphorus limitation. Similar results are found in Doblin et al. (1999) where three marine phytoplankton species (*Gymnodinium catenatum, Alexandrium minutum, Chaetoceros cf.*

*tenuissimus*) showed rapid decline in Se deficient seawater, resulting in cessation of cell division after eight weeks of Se(IV) depletion. However, studies on the effect of selenite in the unicellular green alga *Chlamydomonas*

*reinhardtii*, showed only a little simulative effect on growth (Novoselov et al;

2002). Marine phytoplankton show a stronger trend to a preference for Se than freshwater phytoplankton do (Araie and Shiraiw, 2016).

Geochemical analyses of trace elements in Phanerozoic marine pyrite that sustained periods of severe Se depletion in the past oceans correlate closely with three major mass extinction events, at the end of the Ordovician,

Devonian and Triassic periods (Long et al., 2016). Considering the essential of Se for marine phytoplankton, the authors assumed that Se depletion may have been one of several factors in these complex extinction scenarios (Long et al., 2016).

**Table R1.** Phytoplankton species that were demonstrated to require selenium for their
growth. [a]

| Phylum | Species |
|---|---|
| **Diatoms** | *Amphiprora hyalina* |
| | *Chaetoceros debilis* |
| | *Chaetoceros pelagicus* |
| | *Chaetoceros vixvisibilis* |
| | *Coscinodiscus asteromphalus* |
| | *Corethron criophilum* |

| | *Ditylum brightwellii* |
| | *Skeletonema costatum* (strain 18c NEPCC) |
| | *Skeletonema costatum* (strain 611 NEPCC) |
| | *Skeletonema costatum* (strain 616 NEPCC) |
| | *Stephanopyxis palmeriana* |
| | *Thalassiosira pseudonana* |
| | *Thalassiosira oceanica* |
| | *Thalassiosira rotula* |
| | *Thalassiosira aestivalis* |
| **Dinoflagellates** | *Alexandrium minnutum*[b] |
| | *Gymnodinium catenatum*[b] |
| | *Gymnodinium nagasakiense*[b] |
| | *Peridinium cinctum* fa. *Westii* |
| | *Pyrodinium bahamense*[b] |
| **Prymnesiophytes** | *Chrysochromulina breviturrita* |
| | *Chrysochromulina kappa* |
| | *Chrysochromulina brevefilum* |
| | *Chrysochromulina strobilus* |
| | *Chrysochromulina polylepis*[b] |
| | *Helladosphaera* sp |
| | *Emiliania huxleyi* |
| | *Gephyrocapsa oceanica* |
| **Raphidophytes** | *Chattonella verruculosa*[b] |
| **Chlorophytes** | *Platymonas subcordiformis* |
| **Chrysophytes** | *Aureococcus anophagefferens*[b] |

[a] Modified from Araie and Shiraiw, 2016

[b] Harmful algae.

***Revised manuscripts:***

In *page 2 line 25, "*Selenium (Se) is an essential micronutrient for aquatic organisms".

In *page 3 line 52,* "Se is an essential trace element for aquatic organisms (Bodnar et al, 2014)."

Line 54-67 in the original manuscript "The range of beneficial effects of

Se is among the narrowest of all the elements and varies between dietary deficiency (<40 μg d$^{-1}$) and toxicity (>400 μg d$^{-1}$) (Fernández-Martínez and

Charlet 2009; Schiavon et al., 2017)." were deleted.

In line 63, "marine" have been added before phytoplankton in the revised manuscript, and changed to "A number of field and laboratory studies have found that selenite [Se(IV)] and selenate [Se(VI)] can be assimilated by marine phytoplankton with Se(IV) being the preferred species"

**Comment 2** l. 37: What do you mean by "extremely"? Add numbers.

***Response and revised manuscripts:*** We have changed to "the concentrations of DISe were extremely low (near or below the detection limits, i.e. 0.0063 nmol $L^{-1}$)" in page 2 line 37-38.

**Comment 3** l. 47: I am not sure if the introduction of Se can generally promote productivity. This would only be possible, if Se was an essential nutrient for the considered organism. Furthermore, growth would only be promoted if Se was the limiting element, but other limitations are more likely (e.g., by P or Fe).

***Response:*** The essential requirement for selenium has been reported in

33 species belonging to six phyla (Table R1, Araie and Shiraiw, 2016). The dominant phytoplankton species was diatom in Sarawak coasts (Saifullah et al., 2014). When Se was added to the culture medium, growth was stimulated for diatom, and when it cultures deprived of Se(IV) in seawater for more than

5 days did not recover even when Se(IV) was added afterwards (Harrison et al., 1988). Se-limiting for diatom growth were not found in the filed study, although a study in Huon Estuary found that low level Se could be limiting for growth and production of dinoflagellate (Doblin et al., 1999).

We are not sure whether Se could be a limiting element in the Malaysia coastal waters, thus we have deleted those sentences.

**Comment 4** l. 48-49: I don't understand this conclusion. I would prefer a conclusions, which is derived from your results.

*Response and revised manuscripts:* Thanks. We have deleted the "The results of this study suggest that the impacts of Se discharges on coastal ecosystems should be evaluated in the future", and have changed to "The TDSe flux delivered by the peat-draining rivers exceeded other small rivers, and it is quantitatively more significant than previously thought" in page 2 line 48-50.

**Comment 5.** l. 64: Why is "organic selenide" mentioned separately? It is included in the oxidation state –II.

*Response and revised manuscripts:* We have deleted the "organic selenide" in line 61.

**Comment 6** l. 88: What do you mean by "various", the previously cited studies? Perhaps better cite them again.

*Response:* We have cited the previously mentioned studies again.

*Revised manuscripts:* In page 4 line 88-91:

"Although these various studies did not report different species of Se (Clausen and Brooks, 1983; Lidman et al., 2011; Pokrovsky et al., 2018), the

DOSe probably the dominated species in peatland-draining river"

**Comment 7** l. 94: Do you mean that "Se speciation" was controlled?

*Response and revised manuscripts:* We have changed to "Chang et al. (2016) found that Se speciation was controlled by biological, physical, and redox processes in the estuaries" in page 3 line 74-75.

**Comment 8** l. 97: "formation" or "generation" instead of "regeneration

***Response and revised manuscripts:*** We have changed to "generation of particulate organic selenide in the water." in page 3 line 77.

**Comment 9** l. 106: How does organic matter influence the bioavailability and fate of Se?

***Response:*** We have revised the introduction greatly, added the the character of DOSe in the introduction (see response 11), and "It is also known that organic matter plays an important role in the bioavailability and fate of Se in the environment" in the original manuscripts was deleted.

**Comment 10** l. 110: What do you mean by "behaviour"?

***Response and revised manuscripts:*** We have deleted "the behavior", changed to "More works of Se in fluvial systems in Southeast Asia are therefore needed to provide an improved understanding of the biogeochemical processing of Se and the associations with organic matter." in page 4 line 103-

105.

**Comment 11** l. 122: The third objective "falls from heaven".

***Response:*** Thanks for the great advices. We have revised the introduction, and added the DOSe research (i.e. objective 2 in the revised manuscript) status.

***Revised manuscripts:*** In page 4 line 84-105:

"In the high-latitude peatland-draining rivers, dissolved Se concentrations are spatial variable, with concentrations of up to 13 nmol $L^{-1}$ being observed in northern Minnesota, US (Clausen and Brooks, 1983), from 0.38 to 5 nmol $L^{-1}$

in the Krycklan catchment, Sweden (Lidman et al., 2011) and from 0.25 to 1.25

in the Siberian (Pokrovsky et al., 2018). Although these various studies did not report different species of Se (Clausen and Brooks, 1983; Lidman et al., 2011;

Pokrovsky et al., 2018), the DOSe probably the dominated species in peatland-
draining river. In the open ocean, DOSe was assumed mainly associate with
soluble peptides with low molecule weight in surface waters and were relatively
refractory (Cutter and Cutter, 1995; 2004). Substantial amounts of dissolved Se
also are known to be associated humic substances, Gustafsson and Johnsson
(1994) assumed that Se was preferentially incorporated into low molecular
weight humic substances fractions by means of microbial reductive
incorporation, while Kamei-Ishikawa et al. (2008) found that Se associated with
high molecular weights humic acid fractions. The current paucity of information
on DOSe characteristics and its export by rivers from tropical peat-draining
rivers remains a major gap in our understanding of Se biogeochemical cycling.
Highest concentrations of dissolved organic carbon (DOC) globally were
reported in tropical peat-draining rivers in Borneo (Moore et al., 2013; Wit et al.,
2015). More works of Se in the fluvial systems of this region are therefore
needed to provide an improved understanding of the biogeochemical
processing of Se and the associations with organic matter."

**Comment 12** l. 176: Why did you remove the colloids from the seawater
samples? Doesn't this result in a rather artificial experiment in which some
chemical transformations that can occur in the environment are ruled out?
Please explain. Furthermore, I suggest to come up with a hypothesis, e.g.,
pure mixing vs. chemical transformations (which?).
*Response:* Thanks for the advices. The research aim of this experiment
is to evaluate the impact of particle-free (i.e. dissolved phase) seawater and
river water mixing process on Se species, whether transformation of DISe
between DOSe occurs along salinity gradient. Here, the filter (0.45 micrometer
pore size) could retain a significant portion of colloids while only remove
particles.

The results were shown in Fig. R1. The measured DISe, DOSe and

TDSe concentrations were comparable with theoretical values, indicating pure mixing in Rajang estuary. However, in Maludam estuary, the measured DISe and TDSe concentrations were lower than the theoretical values, while the measured DOSe concentrations were comparable with the theoretical value.

The losses of DISe were not balanced by increasing in DOSe, indicating that chemical transformations between DISe and DOSe did not occur in Maludam.

Other studies have reported removal of the humic fractions of DOM, colloidal iron, and phosphorus by flocculation in the river–sea mixing zones (Eckert and

Sholkovitz, 1976; Forsgren et al., 1996; Asmala et al., 2014). Some of the

DISe may exist in colloidal form in natural water (Takayanagi and Wong,

1984), and DISe may be removed by flocculation. The removal of DISe were probably be flocculated to Se particulate form.

This mixing experiment indeed ruled out the impact of particle and part of colloids on the Se transformations. In the future, we would add another set of mixing experiment without filtration, particle-free and particle-included would be designed to the determine the influence of riverine particles and colloids on

Se chemical cycling. Thus, considering the incompleteness, this section was deleted here.

[Figure]

Fig. R1. Results of laboratory mixing experiments showing variation in DISe, DOSe, and
TDSe concentrations as a function of salinity using filtered riverine water from the Rajang
and Maludam rivers and filtered coastal seawater. TML refers to theoretical mixing line.

**Comment 13.** l. 210: Did you check for normal distribution and transform the data if necessary?

***Response:*** We have checked the normal distribution, and if the data doesn't comply with the normal distribution, Mann Whitney U test were used instead of t-test.

***Revised manuscripts:*** In page 8 line 185-187:

"The Statistical Package for Social Sciences (SPSS) version 23.0 was used to perform Student's t-tests, Mann Whitney U test and linear regression analyses".

**Comment 14** l.214-235: I am a bit lost here. Perhaps, this can be concentrated to the information in aggregated form that is really necessary to understand the results, thereby shortening it.

***Response:*** We have shorted the 3.1 section with main findings from 24

lines to 7 lines.

***Revised manuscripts:*** In page 8 line 190-197:

"The water chemistry in the freshwater reach of the Maludam, Simunjan, and Sebuyau rivers are typical of blackwater rivers draining from peatland with acidic pH and low DO concentrations, and the mixing with coastal water increased the pH and DO (Table S1, Fig. S1). Values of pH and DO

concentrations in the Sematan and Lundu, which drain mostly mineral soils, were higher than those in the blackwater rivers (Fig. S1). In the Rajang estuary, values of pH and DO were lower in the riverine side, especially in the distributaries where covered by the peat (Fig. S2)".

**Comment 15** l 223 and 231: Shouldn't the numbers of the supplementary figures be switched (according to the sequence of their reference in the manuscript)?

***Response:*** We have switched the numbers of the supplementary figures as their sequence in the context.

**Comment 16** 250-268: I suggest to show the results as bar diagram with error and indication of statistically significant difference instead of the current Fig. 2, which I suggest to move to the supplementary information.

***Response and revised manuscripts:*** We have moved the current Fig. 2

to the supplementary information, and draw box plot of TDSe, DISe and

DOSe concentration and DOSe/TDSe ratio in the sampled rivers and estuaries (Fig. 2 in the manuscript).

[Figure]

Figure 2 The box plot of TDSe, DISe and DOSe concentration and DOSe/TDSe ratio in
the sampled rivers and estuaries in Malaysia in March and September 2017, respectively.
In the plot of the upper panel, the ends of the box and the ends of the whiskers, and the
line across each box represent the 25th and 75th percentiles, the fifth and 99th
percentiles, and the median, respectively; the open square indicates the mean value.

**Comment 17** l. 349-358: I suggest to combine the results of each of your three groups into a figure for the group instead of showing all individual results.

***Response and revised manuscripts:*** We have merged Fig. 3 and Fig. 5

to one figure. Three typical groups of relationships between Se species and salinity were selected to present in Fig 3 in the manuscript, including Rajang,

Maludam and Sematan estuaries, and those for Sebuyau and Samunsam were moved to supplement (Fig. S5 in the supplementary).

[Figure]

Fig. 3 Relationships between DISe (a - d), DOSe (e - h), and TDSe (i - l) concentrations
with salinity in the Rajang and three Rajang tributaries (Igan, Lassa, and Rajang), and in
the Maludam and Sematan estuaries in March and September 2017. TML refers to the
theoretical mixing line, which was defined using two endmembers: freshwater in the
riverine system and seawater.

**Comment 18.** l. 384ff: The Discussion section should be structured according to the three objectives into three parts. The objectives should be discussed as concisely as possible, i.e. the current discussion should be shortened.

***Response:*** We have revised the objectives and reorganized the discussions according to the objectives. The discussion session was shorted from 280 lines to 152 lines.

The three objectives were 1) evaluate the fate of Se species during estuarine mixing in peatland-draining estuaries; 2) characterize the DOSe fractions; and 3) estimate the magnitude of Se fluxes delivered from peatland- draining rivers to coastal ocean. The discussion and conclusion sections were structured to three parts according to the objectives, as following: 4.1) Fate of

Se species during estuarine mixing; 4.2) Character of the DOSe fractions; and

4.3) TDSe flux.

**Comment 19** l. 407-412: This belongs to the results.

*Response:* We have moved those to the results as section. "3.4

Correlation between Se species with DO, pH and DOM".

*Revised manuscripts:* In page 11 line 274-287:

"For the freshwaters (S < 1) of the studied rivers, DISe concentrations were positively correlated with the DO concentrations and pH values, and the

DISe/DOSe ratio was negatively related to DOC concentration (data from

Martin et al., 2018) (Fig 4a, 4b); DOSe concentrations correlated positively with the humification index (HIX) and the sum of the humic-like chromophoric dissolved organic matter (CDOM components, C1, C2, C3, and C4) ($p < 0.05$)

(data from Zhou et al., 2019) (Fig 4c, 4d).

In the Maludam Estuary, DOSe concentrations were negatively correlated with the CDOM spectral slope from 275 to 295 nm ($S_{275-295}$) and were positively correlated with the humic-like C3 component and specific UV absorbance at

254 nm ($SUVA_{254}$) during estuarine mixing in both seasons (data from Martin et al., 2018; Zhou et al., 2019) (Fig 5a-c). In addition, DOSe/DOC and DOSe/DISe ratios were negatively correlated with C2/C1 components ratios (Fig 5d; 5e)."

[Figure]

Figure 4. Relationships between (a, b) DISe concentrations and DO and pH values, (c)
DISe/DOSe ratios and DOC concentration values, and (d–e) DOSe concentrations with
the humification index (HIX) and the sum of humic-like CDOM components (C1, C2, C3,
and C4) in freshwater (Salinity < 1) for the Rajang, Sematan, Maludam, Sebuyau,
Samunsam, and Simunjan rivers in March and September. The HIX and C1, C2, C3, and
C4 components are from Zhou et al. (2019) from the same cruises. DO concentrations and
pH values were not available for the Sematan River for September, and the HIX and CDOM
components were not available for the Rajang River for September.

[Figure]

Figure 5. Relationships between DOSe concentrations and $S_{275-295}$, C3 components and
$SUVA_{254}$, DOSe/DOC ratio and C2/C1 component ratios, and DOSe/DISe ratios and C2/C1
component ratios in the Rajang and Maludam estuaries. The $S_{275-295}$, $SUVA_{254}$, C1, C2,
and C3 components are from Martin et al. (2018) and Zhou et al. (2019) from the same
cruises.

**Comment 20** l. 432-433:Why is this information important? I suggest to delete it.

*Response:* We have deleted the "Se sorption kinetics on humic acids can be expressed by a pseudo-second-order equation (Kamei-Ishikawa et al., 2007)."

**Comment 21** l434-441: This is a repetition. Delete and focus on the important statement in l. 441-442.

*Response:* We have deleted the "The Maludam, Sebuyau, and Simunjan catchments are mainly peat, whereas the Samunsam River drains an extensive area of peatland in its upper reaches (Müller et al., 2016; Martin et al., 2018). The Rajang catchment is dominated by mineral soils, with peatland being found only in the delta surrounding the distributaries (Staub et al., 1994, 2000)"

**Comment 22** l. 469: You can omit "as follows" and directly start with the numbered list.

*Response:* We have deleted the "as follows"

**Comment 23** l. 479: Combine 4.1 and 4.2 as joint contribution to Objective 1.

*Response:* Thanks for the great advice. We have combined the 4.1 and 4.2 to "4.1 Fate of Se species during estuarine mixing", as joint of contribution to Objective 1.

*Revised manuscripts:* In page 12-14, line 289-364:

**"4.1 Fate of Se species during estuarine mixing**

[revised manuscript text omitted]

**Comment 24** l. 541/545: Isn't the Rhone forming a delta? Or don't you talk about the French Rhone?

*Response:* Thanks, it's the French Rhone delta.

**Comment 25** l. 549: Skip this heading to avoid overstructuring.

*Response:* We have reorganized the discussion, skip the heading that mentioned.

**Comment 26** l. 562: I am confused by the simultaneous use of delta and estuary, because I think that these are two contrasting geomorphological forms, mainly driven by the strength of the tide.

*Response:* We have unified the expression of estuary instead of delta in the manuscripts to avoid confusion.

**Comment 27** l. 563-573: I would move large parts of this and the associated figure to the results section.

*Response:* We have moved those to the results as section. "3.4

Correlation between Se species with DO, pH and DOM", detailed response was shown in comment 19.

**Comment 28** l. 588-589: The numbers should be subscripts.

*Response:* In the FDOM area, the C1 represents Component 1 that decomposed from the excitation-emission matrix. In the previous publications,

C1 and C2 were widely used. For example, Table 1 in Osburn et al. (2012),

Table 1 and Figure 2 in Dainard et al. (2015), there is no need to be subscripts.

**Comment 29** l. 593-601: I would again move large parts of this and the associated figure to the results section.

*Response:* We have moved those to the results as section. "3.4

Correlation between Se species with DO, pH and DOM", detailed response was shown in comment 19*.*

**Comment 30.** l. 622: Will photodegradation really be important in the dark

DOM-rich waters? Possibly, it is restricted to the uppermost surface-near few mm.

*Response:* Martin et al (2018) found that DOM from the Rajang and

Samunsam rivers was photolabile, with DOC and CDOM decreasing after sunlight exposure. In addition, the peatland-derived DOM probably has too short a residence time in rivers for significant photodegradation to occur in the rivers before it reaches the sea, thus the authors suggested that most photochemical transformations of tDOC in Sarawak likely take place after tDOC reaches the sea rather than inside the rivers and estuaries.

*Revised manuscripts:* In page 15, line 392-398:

"As suggested by Martin et al (2018) that most photochemical transformations of DOM in Sarawak likely take place after DOM reaches the sea. Thus, once transported to offshore, photodegradation plays an important role in DOSe processing, and DOSe might contain a significant photoreactive fraction that facilitates photodegradation of DOSe into lower mean molecular weights or gaseous Se or photomineralization to DISe."

**Comment 31** l. 665-678: The conclusions should fit to the objectives, i.e.

there should be three main conclusions and perhaps a kind of outlook.

*Response:* Thanks for the great advices. We have revised the three objectives, were 1) evaluate the fate of Se species during estuarine mixing in peatland- draining estuaries; 2) characterize the DOSe fractions; and 3) estimate the magnitude of Se fluxes delivered from peatland-draining rivers to coastal ocean.

The conclusions were reorganized to fit the objectives and have an outlook.

*Revised manuscripts:* In page 17, line 440-455:

"To the best of our knowledge, this is the first study of seasonal variations in Se speciation in peat-draining rivers and estuaries in Southeast Asia.

Contrary to the results from studies elsewhere, DOSe, not DISe, was the major species in the peat-draining rivers and estuaries of Sarawak, Malaysia.

Contrary to our expectations, reversed DISe concentration–salinity relationships were observed in those estuaries, indicating a marine origin, while

DOSe concentrations decreased with salinity, indicating terrestrial sources. The

DOSe fractions may be associated with high-molecular-weight peatland- derived aromatic and black carbon compounds and may photodegrade to more bioavailable forms once transported to oligotrophic coastal waters, where they may stimulate the growth of phytoplankton. The DOSe yields in the peatland- draining rivers were one or even two orders of magnitude higher than other reported rivers. The TDSe flux delivered by the exceeded other small rivers, and it is quantitatively more significant than previously thought. The impact of the sizable Se from increasing anthropogenic disturbing of peat to the ecosystem should be evaluated in the future"

**Comment 32**. l 678: See my previous comments to the role of Se for biological productivity.

*Response:* The detailed response was shown in comment 3. We have changed to "may stimulate the growth of phytoplankton" in line 450.

**Comment 33** Figs. 3-5 show all individual results. I suggest to aggregate these data in a way that clearly illustrates your main points.

*Response:* We have moved Fig. 2 and Fig. 4 to the supplement, and Fig.

3 and Fig. 5 were merged to one figure. Three groups of the relationships between Se species and salinity were selected to present in Fig. 3, including

Rajang, Maludam and Sematan estuaries, and those for Sebuyau and

Samunsam were moved to supplement.

*Revised manuscripts:* Figure 3 were present in *page 29 line 738-743.*

**Comment 34** Figs. 7 and 8 should be included in the Results section.

*Response:* We have moved those figures to the results in section. "3.4

Correlation between Se species with DO, pH and DOM". The detailed response was shown in comment 19, and figures were show as Fig. 4 and Fig

5.

*Revised manuscripts:* Figure 3 were present in *page 29 line 738-743.*

---

## Author Response (AR2)

**Response to review**

**Respect Dr. Bouillon and Dr. Pokrovsky:**

We would like to thank Biogeosciences for accepting our manuscript.

We thank the reviewer and editor for their valuable comments on revised manuscript. Based on the comments, we have made carefully revision.

**Editor's Comment:**

**Comment 1.** L38 and L 218: detection limits: detection limit

*Response:* Thanks. We have changed to detection limit in the revised manuscripts.

*Revised manuscripts: "*the concentrations of DISe were extremely low (near or below the detection limit, i.e. 0.0063 nmol L$^{-1}$)" in *page 2 line 37-38.*

"DISe concentrations were extremely low (near or below the detection limit) in the freshwater reach and increased towards the sea*" in *page 9 line 217-219.*

**Comment 2.** L48: "The TDSe flux delivered by the peat-draining rivers exceeded other small rivers reported so far": exceeded those reported for other small rivers ?

*Revised manuscripts: "*The TDSe flux delivered by the peat-draining rivers exceeded those reported for other small rivers" in *page 2 line 48-49.*

**Comment 3.** L66: "in diatom": in diatoms

***Revised manuscripts:*** *"*"Direct uptake of seleno-methionine and seleno- cystine has been demonstrated ==in diatoms==" in *page 3 line 65-66.*

**Comment 4.** L88: "in the Siberian": awkward, rephrase or be more specific

***Revised manuscripts:*** *"*In the high-latitude peatland-draining rivers, dissolved Se concentrations are spatially variable, with concentrations of up to

13 nmol $L^{-1}$ being observed in northern Minnesota, US (Clausen and Brooks,

1983), from 0.38 to 5 nmol $L^{-1}$ in the Krycklan catchment, Sweden (Lidman et al., 2011) and from 0.25 to 1.25 nmol $L^{-1}$ ==in the lakes and rivers of western==

==Siberian==" in *page 4 line 84-88.*

**Comment 5.** L191: "draining from peatland": draining peatlands

***Revised manuscripts:*** *"*The water chemistry in the freshwater reach of the Maludam, Simunjan, Sebuyau and Samunsam rivers are typical of blackwater rivers ==draining peatlands== with acidic pH and low DO

concentrations" in *page 8 line 190-192.*

**Comment 6.** L270: "were removed in the brackish water": in the brackish water region

***Revised manuscripts:*** *"*DISe increased with salinity but behaved non- conservatively and was removed in the brackish water ==region==" in *page 11 line*

*269-270.*

**Comment 7.** L305: "be expected to": are expected to

***Revised manuscripts:*** *"*thus DISe concentrations ==are expected to==

increase with DO values" in *page 12 line 304-305.*

**Comment 8.** L323: "which were contrast with": which contrasts with, or: which is in contrast with

*Revised manuscripts:* "During estuarine mixing, reversed DISe concentration–salinity relationships were observed in the Rajang, Maludam, Sebuyau, and Samunsam estuaries (Fig. 3, Fig S5), which ==contrasts== with those reported for other estuaries" in *page 13 line 321-324.*

**Comment 9.** L373: "liner": linear

*Revised manuscripts:* "Moreover the peat-draining rivers demonstrated a ==linear== relationship between DOSe concentrations and HIX and humic-like CDOM components (Fig. 4d, 4e) indicating that DOSe may be associated with dissolved humic substances." in *page 15 line 375-378.*

**Comment 10.** L379-380: "the humic-like C3 component (Fig. 5b) which derived corresponded to aromatic and black carbon compounds": something wrong with this sentence, rephrase.

*Revised manuscripts:* "In addition, the positive correlations between DOSe and the humic-like C3 component (Fig. 5b), i.e. aromatic and black carbon compounds, suggest a strong association of DOSe to these high molecular weight DOM" in *page 15 line 378-381.*

**Comment 11.** L382-383: "Pokrovsky et al. (2018) also found that Se were transport in the form of …": was transported in the form of

*Revised manuscripts:* "Pokrovsky et al. (2018) also found that Se ==was== transported in the form of high molecular weights organic aromatic-rich complexes from peat to the rivers and lakes in the Arctic." in *page 15 line 381-383.*

**Comment 12.** L386: remove "that"

*Revised manuscripts:* "Bruggeman et al. (2007) and Kamei-Ishikawa et al. (2008) both found that 50% to 70% of Se(IV)–humic substances associates had high molecular weights (>10 kDa), consistent with our findings. " in *page 15 line 383-386.*

**Comment 13.** L426: "were exceed the other rivers": but exceeded the other rivers

*Revised manuscripts:* "The TDSe yields for Rajang and Maludam were just below the second largest river Changjiang and the polluted Scheldt River, but exceed the other rivers" in *page 17 line 424-426.*

**Comment 14.** L427: "As for ..": awkward sentence, rephrase

*Revised manuscripts:* "The magnitude of DOSe yields obtained from Rajang and Maludam was one to two orders of degree higher than those in other reported rivers" in *page 17 line 426-428*

**Comment 15.** L452: "The TDSe flux delivered by the exceeded other small rivers": something missing here: delivered by what ?

*Revised manuscripts:* "The TDSe flux delivered by the peat-draining rivers exceeded other small rivers" in *page 17 line 450-451.*

**Comment 16.** Table 1: 'coverage rate': this is not a rate, rephrase to for example 'peatland cover (%)'

*Response:* We have changed to "Peatland cover" in *page 26 line 710.*

**Reviewer's comments:**

**Comment 1.** L85 spatially

*Revised manuscripts:* "In the high-latitude peatland-draining rivers, dissolved Se concentrations are spatially variable, with concentrations of up to 13 nmol $L^{-1}$ being observed in northern Minnesota, US (Clausen and Brooks, 1983), from 0.38 to 5 nmol $L^{-1}$ in the Krycklan catchment, Sweden (Lidman et al., 2011) and from 0.25 to 1.25 nmol $L^{-1}$ in the lakes and rivers of western Siberian" in *page 4 line 84-88.*

**Comment 2.** L90 the DOSe is probably

*Revised manuscripts:* "the DOSe is probably the dominated species in peatland-draining river" in *page 4 line 90-91.*

**Comment 3.** L99 its export by tropical peat-draining…

*Revised manuscripts:* "The current paucity of information on DOSe characteristics and its export by tropical peat-draining rivers remains a major gap in our understanding of Se biogeochemical cycling" in *page 4 line 99-101.*

**Comment 4.** L383 transported in the form

*Revised manuscripts:* "Pokrovsky et al. (2018) also found that Se was transported in the form of high molecular weights organic aromatic-rich complexes from peat to the rivers and lakes in the Arctic" in *page 15 line 381-383.*

**Comment 5.** L432 remove one 'roughly'

*Revised manuscripts:* "The roughly estimated TDSe flux from tropical peatland (439,238 $km^2$, Page et al., 2011) could be around $120 \times 10^3$ kg $yr^{-1}$" in *page 17 line 430-431.*

**Comment 6.**L435-438 Rephrase or split into 2 sentences

*Revised manuscripts: "*It can be expected that increasing anthropogenic disturbing of peat can release a great amount of Se to rivers, and then transported to the coastal areas. The impact of peatland derived Se to the ecosystem should receive more attention in future studies." in *page 17 line*

*435-438.*

**Comment 7.**L447-450: Photodegradation and plankton growth were not investigated in this study so it should not be in the Conclusions

*Response:* Photodegradation and plankton growth were deleted.

*Revised manuscripts: "*The DOSe fractions may be associated with high- molecular-weight peatland-derived aromatic and black carbon compounds." in

*page 18 line 466-468.*